# Heron, a Knowledge Graph editor for intuitive implementation of Python-based experimental pipelines

George Dimitriadis[1]*, Ella Svahn[1,2], Andrew F MacAskill[2], Athena Akrami[1]

[1]Sainsbury Wellcome Centre, University College London, London, United Kingdom; [2]Department of Neuroscience, Physiology and Pharmacology, University College London, London, United Kingdom

## eLife Assessment

This **valuable** paper introduces Heron, lightweight scientific software that is designed to streamline the implementation of complex experimental pipelines. The software is tailored for workflows that require coordinating many logical steps across interconnected hardware components with heterogeneous computing environments. The authors **convincingly** demonstrate Heron's utility and effectiveness in the context of behavioral experiments, addressing a growing need among experimentalists for flexible and scalable solutions that accommodate diverse and evolving hardware requirements.

**\*For correspondence:**
g.dimitriadis@ucl.ac.uk

**Abstract** To realise a research project, experimenters face conflicting design and implementation choices across hardware and software. These include balancing ease of implementation – time, expertise, and resources – against future flexibility, the number of opaque (black box) components and reproducibility. To address this, we present Heron, a Python-based platform for constructing and running experimental and data analysis pipelines. Heron allows researchers to design experiments according to their own mental schemata, represented as a Knowledge Graph – a structure that mirrors the logical flow of an experiment. This approach speeds up implementation (and subsequent updates), while minimising black box components, increasing transparency and reproducibility. Heron supports the integration of software and hardware combinations that are otherwise too complex or costly, making it especially useful in experimental sciences with a large number of interconnected components such as robotics, neuroscience, behavioural sciences, physics, chemistry, and environmental sciences. Unlike visual-only tools, Heron combines full control (of instrument and software combinations) and flexibility with the ease of high-level programming and Graphical User Interfaces. It assumes intermediate Python proficiency and offers a clean, modular code base that encourages documentation and reuse. By removing inaccessible technical barriers, Heron enables researchers without formal engineering backgrounds to construct sophisticated, reliable and reproducible experimental setups – bridging the gap between scientific creativity and technical implementation.

## Introduction

One can divide an experiment's life circle, from concept to a running system, into a number of transformations. At first, the scientific question is mapped into an abstract schema of experimental steps (i.e. what needs to happen in order to answer the question). Subsequently, these conceptual steps are transformed into a schema of hardware connectivity and software logic. At this stage, the experimenter thinks in terms of high-level objects like cameras and other types of sensors, time lines,

**eLife digest** Complex scientific experiments often require setting up several pieces of hardware and software that must work together seamlessly. It is crucial not only to connect these components correctly, but also to design the setup in a way that others can easily replicate and adapt for similar experiments.

Striking a balance between a system that works correctly and one that can be quickly adjusted and understood by other users can be difficult. This is especially relevant when systems incorporating complicated software and hardware are to be designed and used by scientists who may not have expertise in those areas.

Dimitriadis et al. set out to create a software tool that would help researchers of all disciplines build and run complex experiments more easily. The resulting platform, known as Heron, lets users create setups by combining visual building blocks representing parts of an experiment. These blocks are arranged in what is called a Knowledge Graph, which shows how different steps in the experiment connect in a way that closely mirrors the thought process of the researcher.

This approach makes experiments quicker to set up, easier to update, and more transparent for others to replicate or understand, especially in fields like robotics or neuroscience where complex setups are common. It also results in code that is easier to understand, maintain and share with others. These factors will help Heron to enhance how reproducible experimental setups are and allow researchers to use combinations of hardware and software that would be difficult to achieve otherwise.

triggers and events, agents and rewards, inputs and outputs. The final challenging step, which is rarely addressed (or even cognitively acknowledged), is to map the schema of the hardware and software logic to the actual hardware connectivity and operational code bases. At this level, the experimenter has to work with much lower-level objects like voltage differences, light intensities, TTL pulses, GPU shaders, and information flow loops. This last and most time-consuming step can limit the number of iterations for ideas to be piloted and tested. Once the mental schema has been translated into code, it is this code that is usually addressed by other experimentalists in reproducibility efforts. These efforts usually face a high barrier in understanding the original experimental schema starting from its code implementation. This is because efficiently translating many lines of code (even if well documented) back to what the code actually accomplishes is in most cases a difficult task that requires years of coding experience, as well as familiarity with experimental designs. This barrier hinders efforts of reproducibility and quality control and is one that cannot be addressed solely by open sourcing one's work. Finally, the complexity of the translation from mental schema to running code base makes design iteration efforts practically impossible. The prohibitively large iteration time, on the one hand, and the inter-dependency of engineering decisions throughout the implementation cycle, on the other hand, make updates of the experimental flow extremely cumbersome. This often results in practically one of the most serious experiment design and implementation hurdles. The need to create radically new implementations for only small changes in the underlying mental schema. These arguments can be seen as the main driver for the development and the rapid acceptance of high-level languages in software engineering and of micro-controller kits (e.g. Arduino) in electrical engineering and robotics. Concepts like object-oriented programming (*Kindler and Krivy, 2011*) or actor-based programming (*Agha, 1986*) for example, have been nothing else but an effort to take away the low-level concepts that one needs to ultimately manipulate and replace them with bundles of higher-level ones, easier to cognise with.

In order to address this discrepancy between the mental schemata and implementation outcomes in experimental construction, we developed Heron, an open source software (MIT license) platform for the construction of data flow pipelines (e.g. experiments, data analysis, robotics, etc.). Heron comes with a series of distinguishing features. The primary one is that it creates experimental pipelines that visually and structurally bear a very significant resemblance to the original mental schema of the experimental pipeline. So what one gets as the final experiment implementation is both semantically and syntactically very close to how one originally envisions the experiment should work. Because of that, Heron creates final implementations that are easy to understand, construct, communicate, and change. In this way, it often makes it fairly easy to put together helpful diagrams and documentation

by following Heron's visual representation of the experiment and its underlying code. It also allows for accurate inference of the real-time complexity of any proposed change even before any new code is written.

A second distinguishing feature of Heron is that it targets the experimenter without expertise in arcane subjects like networking, hardware connectivity, or low-level software–hardware interactions. By abstracting away these low-level features in its Graphical User Interface (GUI), it allows the construction of experiments with multiple, diverse hardware components, even using networks of computers running different operating systems, without bothering the experimenter with having to deal with all the low-level connectivity issues that arise. It achieves this without limiting its users to preconceived ideas of how any single specific piece of hardware or code should be used. This is possible as Heron offers users an Application Programming Interface – API – letting them write the code that implements their own ideas at the optimal level of abstraction given the situation. Offering Python as the main (but not only) language for this user-centric code implementation, Heron makes it even easier for code novices to achieve highly complex experimental setups that are easy to both construct and reconfigure. In the following section, we will focus on Heron as a general-purpose tool for constructing pipelines used to conduct different types of experiments. Firstly, we will describe the specific meaning of an experimental pipeline implemented as a Knowledge Graph (KG) (*Fensel et al., 2020*). We then catalogue the design benefits and distinguishing features offered by Heron, in comparison to other efforts targeting the construction of experiments. Subsequently, we will describe the internal architecture of Heron with enough details to allow any developer to quickly get up to speed with Heron's code and contribute to its open source. Finally, we will illustrate a number of Heron experimental implementations, currently in use in the lab, each showcasing one of Heron's special features.

## Results

### Mental schemata, KGs and Heron processes. The philosophy behind Heron's design

A mental schema is a psychological concept (*Bartlett and Burt, 1933*, *Williams, 2019*) that is meant to define the way humans cognise. According to it, when people think, they categorise sensory experiences and abstract notions into groups. They then utilise the relations between these groups to draw conclusions about some hypotheses. Upon inputs from the environment and prior cognitive outcomes, the categories and their relations can update fluidly (*Williams, 2019*). An example, for the case of interest here, is the mental schema of an experiment. In order for an experiment to be developed, the experimenter brings together a set of categories, both based on their sensory experiences (e.g. a laser, a data acquisition board, or a camera) and on abstract notions (e.g. time concurrency or subject's choices). Then a set of relationships is generated between them. The sets of these concepts with their interactions define the mental schema of the specific experiment. For example, a camera frame must be captured immediately after a specific event has taken place. Yet, today's implementations of experimental pipelines are written (with visual or text-based code) such that they obfuscate the mental schemata they derive from. A receiver of such a code base, irrespective of their understanding of the underlying language, always needs a significant amount of time and mental effort to map back the initial mental schema. A KG (*Fensel et al., 2020*) is a mathematical structure designed to capture the unstructured human knowledge on a subject (i.e. the mental schemata of different individuals relating to the same knowledge corpus) in such a way that a machine could use it to test propositions against the knowledge and also create novel propositions that a human with the knowledge would find to be true. It is practically an effort to implement the fuzzy notion of a mental schema into a concrete structure of objects and relations that is machine implementable. The fundamental structures of a KG are its nodes and their attributes. Nodes are meant to represent a group of objects at a desired level of abstraction, which does not have to be uniform among the different nodes of the same KG. Nodes' attributes define the state of each node and the edges that connect different nodes representing their relationships. Nodes and their attributes usually have semantic labels.

Heron implements an experiment in the form of a KG and does so at two separate levels. One is the graphical level, where a series of Nodes and their in-between links (Edges) are defined. The second is the code that defines each Node's functionality. The graphical level is used to construct a KG

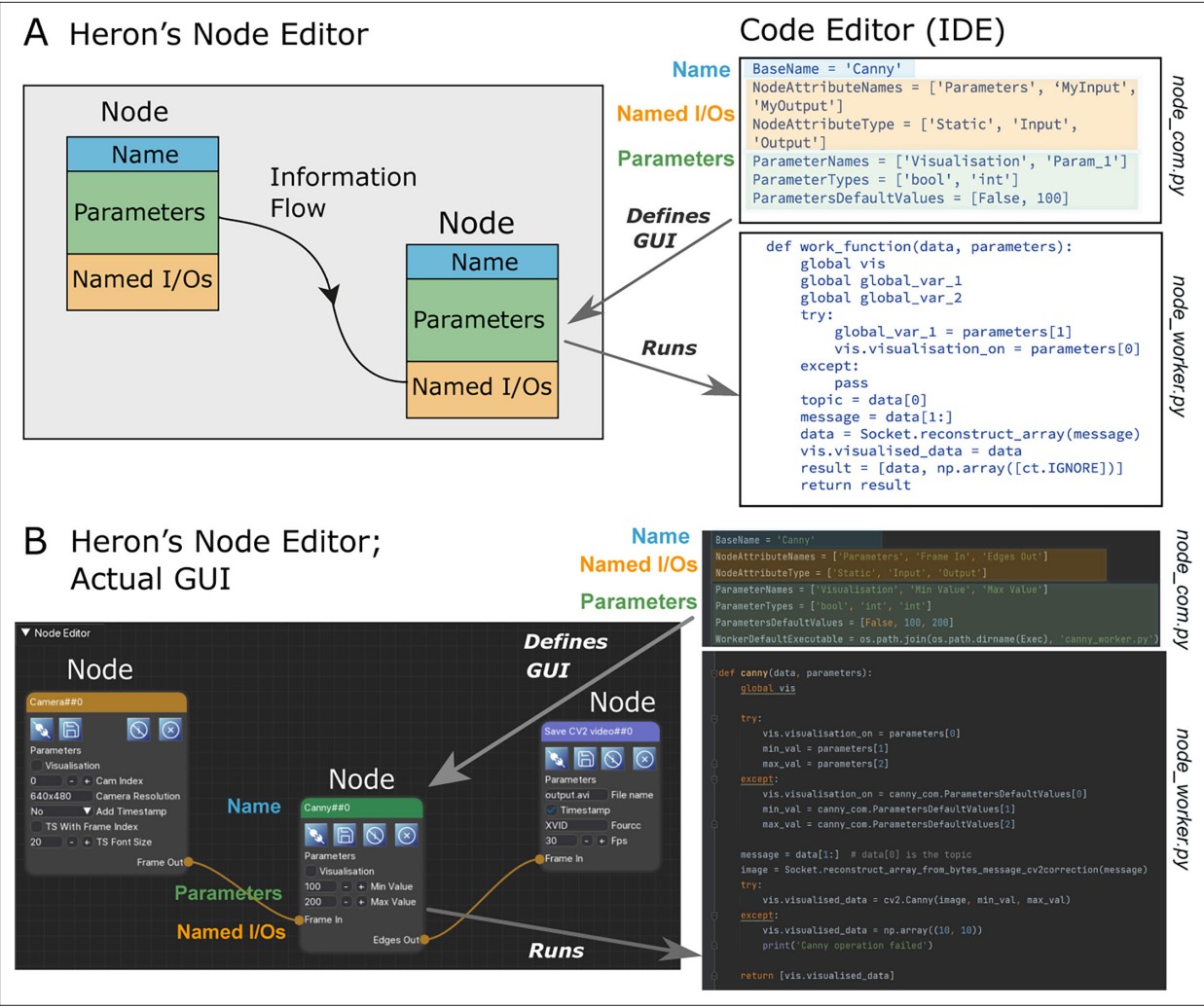

**Figure 1.** Heron's combination of graphical and text-based code development. (**A**) Schematic of Heron's two levels of operation. Left shows a Knowledge Graph made of Nodes with Names, named Parameters and named Input/Output points. Nodes are connected with links (Edges) that define the flow of information. Right shows a piece of textual code that fully defines both the Node's Graphical User Interface (GUI) appearance and the processes run by the machine. (**B**) Heron's implementation of A, where the Knowledge Graph is generated in Heron's Node Editor window while the code that each Node needs to be defined and the code that each Node is running is written in a separate Integrated Development Environment (IDE, in this case PyCharm).

representing the experiment. The text-based code level is used to implement the dynamics of each Node in the Graph and define its level of abstraction. The KG's Nodes are labelled and have human-readable attributes partially defining their state. They are also connected with directed Edges (links) through their named input and output points. The Edges represent message passing between Nodes (see *Figure 1*). In this way, a graphical representation of an experimenter's mental schema is created with each Node at the appropriate level of abstraction for the specific experiment. This individual-ised level of abstraction is achieved through Heron's second level of implementation, that is a text-based code that defines each Node's functionality. This is an important distinction between Heron and other node-based software like LabView and Bonsai (see Background section). Heron expects that each Node's behaviour and connectivity is defined by the experimenter, in normal, text-based code, and comes with an appropriate API to facilitate this. In this way, Heron does not offer a long list of predefined Nodes which would make very strong assumptions about the structure of the experi-ment's mental schema. Instead, it offers the tools for designing and implementing one's own Nodes in a case-by-case approach at a level of abstraction required at any time. For instance, if in one case a Node representing a camera is required, while in another a Node should represent a large group of

synchronised cameras acting as one, Heron provides the tooling to create either of these Nodes with minimum effort.

Heron's code base is implemented in Python. This choice was informed by the need to keep the code simple to implement while at the same time being able to interact with a very diverse list of hardware drivers and software analysis libraries (see Design benefits section). It also means that Python is the go-to language for the user's creation of new Node behaviours. Yet the language that Heron itself is implemented in (Python) does not enforce the language new Nodes can be implemented in by Heron users. Heron offers users a comfortable starting point to implement their own Nodes (in Python), while allowing for expert users to utilise lower-level languages that offer other advantages like faster run times and lower-level control of a machine's components (e.g. CPUs, GPUs, RAM, etc.). Heron poses very few (if any) limitations in what types of behaviour can be implemented in any Node. Breaking up the implementation into two levels of KG and running codes, as mentioned above, confers a much-required 'break complex problems into simpler ones' functionality at the heart of Heron's operation.

## Background

### Computational graphs

Heron's KG approach has its root in a number of software frameworks where experimental and data analysis pipelines are constructed as a computational graph. MATLAB Simulink (*Acklam, 2003*) and LabVIEW (*Bitter and Nawrocki, 2006*) are two of the more well-known frameworks where experiments can be designed as computational graphs using predefined elements (i.e. nodes provided by the developers of these systems). Bonsai (*Lopes et al., 2015*) is a new entry to this field originally designed to ease the implementation of neuroscience and behavioural science experiments. The use of directed acyclic graphs (*Thulasiraman and Swamy, 1992*) has increased dramatically in the Big Data analytics tools where frameworks like Apache Airflow (*Apache, 2015*) and Dask (*Rocklin, 2015*) allow parallelisation of algorithms and data queries over large clusters of machines. Two efforts that are very similar to Heron in the way they structure computational graphs to define experimental pipelines are EPypes (*Semeniuta and Falkman, 2019*) and the Robot Operating System (ROS) (*Stanford Artificial Intelligence Laboratory, 2018*). EPypes is (according to the developers' own description) a 'Python-based software framework for developing vision algorithms in a form of computational graphs and their integration with distributed systems based on publish-subscribe communication'. The basic idea of message passing between individual processes, each responsible for its own algorithm, running on different machines is identical to Heron (even at the level of using the publish–subscribe communication protocol), although EPypes's focus is on computer vision algorithms. The exact same idea is utilised by ROS where (in their own words again) 'The ROS runtime 'graph' is a peer-to-peer network of processes (potentially distributed across machines) that are loosely coupled using the ROS communication infrastructure. ROS implements several different styles of communication, including synchronous RPC-style communication over services, asynchronous streaming of data over topics, and storage of data on a Parameter Server'.

### Hybrid programming

Heron's approach is based on an existing programming idea, that is combining both visual and textual programming in a hybrid manner. One of the most widely used examples of such realisations is VVVV (*Holzer, 1998*), a framework utilising visual programming with the C# or HLSL programming languages for textual programming. This hybrid approach was found to allow for a better retention of computer software university students, when comparing only textual or only visual styles in the learning of programming (*Noone and Mooney, 2018*), showing that it better suits beginner level programmers (a category into which a large percentage of experimentalists fall in).

### Behavioural sciences toolboxes

Heron was originally conceptualised to be a framework for creating experiments in the fields of behavioural sciences (e.g. neuroscience, experimental psychology, etc.) and although its philosophy and use cases span a much wider spectrum, its current usage derives from experiments in this field. Other frameworks that specifically target the same fields are pyControl (*Akam et al., 2022*), Bpod (Sanworks LLC, USA) (building on the central design concept of B-control; *Brody, 2007*), and

Autopilot (*Saunders and Wehr, 2019*). Bpod and pyControl are software-dedicated hardware efforts, while Autopilot is a software framework that, in the same spirit as EPypes, ROS and Heron, allows a distributed experimental pipeline, albeit restricting the machines to Raspberry Pi computers. All three efforts pay special attention to offering their users tools for creating state machines to define their experiments (each utilising its own way of doing so). Heron currently allows the users to decide if their pipeline would benefit from a state machine design or not, and being Python-based allows for the use of a plethora of state machine tools in the Python ecosystem (*Macedo, 2025*; *Sivji, 2020*). This includes the capability to script Nodes that can wrap the Python APIs of pyControl and Bpod (through the Champalimaud Foundation's pybpod API). This can be of interest to those experimentalists who have invested in the respective hardware modules but would like to expand the capabilities of their pipelines beyond the reach of pyControl or Bpod.

## Design benefits

### Self-documentation

The KG of Heron immediately offers a succinct overview of the experimental workflow and the dynamics it implements, thus acting also as the primary documentation of the experiment. Armed with a coherent picture of the experiment's information flow over time, one can access the code of individual Nodes, for a deeper understanding of its details. Grasping the meaning of a few hundred lines of Python code that most Nodes require to be implemented, one Node at a time, is a much more appealing proposition than opening up a whole code base of a non-Heron experiment and being faced with thousands of lines of obscurely interconnected code arranged in a file system that only makes sense to the developer (and only for a short while after the code's implementation). Moreover, Python code, in comparison to other lower-level languages, helps with readability in a self-documenting fashion (notwithstanding the plethora of in-code documentation tools in the Python ecosystem). This self-documentation capability of Heron's experimental implementations confers obvious benefits to the exchange and reproducibility of experiments and minimises the possibility of misunderstandings when researchers from different groups try to interact with the experiment.

### Multiple machines, operating systems, and environments

In Heron, each Node runs its own process (practically its own little program, separate from all the programs of the other Nodes). This multi-process approach offers an important competence; running different Nodes on different machines (albeit by taking a hit on system resources vs a multi-threaded approach). This is important since experiments should not be constrained by the Operating System (OS) or the chip architecture that a small part of the experiment might require to run. For example, a fast, high-resolution camera might have drivers only for Windows, while raspberry pi cameras can be advantageous since they are easy to multiplex (due to the pi's GPIO and low cost of a raspberry pi with a camera) while online, million-parameter, deep learning algorithms will definitely not run on anything other than high spec Linux machines. Heron removes the need to choose between these capabilities since its Nodes (i.e. processes) can run on any machine connected to the main one that runs Heron, through a Secure Shell Protocol (SSH) accessible network connection. When a KG is initiated (i.e. a task is launched), Heron will connect to all the defined machines in the network and will initialise whatever processes it has been directed to start at each of its predefined machines. While the experiment is running, it will take care of message passing between machines, and when the Graph is terminated, it will make sure all processes are also gracefully terminated. Since Heron uses standard Python to implement most of its Nodes (something that users, as we mentioned above, do not have to adhere to since functionality to work with other code bases exists – see the Rats playing computer games: State machines and non-Python code Nodes example), a Heron experiment can be easily defined on machines with different chip architectures, different OSes and different levels of virtualisation. The general rule is that if a machine with a certain configuration can run the scripts of Python (or other non-Python code) that define the Nodes that need to run on that machine, then that script can be part of a Heron experiment and have its Node's inputs and outputs interact with Nodes running on other machines, all set up through Heron's GUI and with minimal user effort. Heron itself (the GUI and underlying communication functionality) has been tested to be operational under Windows (10 and 11, both ×64), Linux PC (Ubuntu 20.04.6, ×64) MacOS, and Raspberry Pi 4 (Debian GNU/Linux 12 -bookworm-, aarch64).

## Python and the ease of implementing code

Finally, Python as an implementation language offers Heron another set of desirable (and some not so) consequences. These include the standard pros and cons of Python versus other computer languages. Apart from this, batteries included, Python's approach to problems, the main advantage for experiment implementation is Python's extensive community of developers that have contributed to the open source ecosystem of Python libraries for practically any computation imaginable. This extends to drivers and control APIs for most hardware that an experimenter might require to use. From standard data crunching algorithms to state-of-the-art machine learning ones and from serial communication to drivers controlling high-spec equipment, there is very little that has not been covered by a Python library. This wealth of ready-to-go solutions makes the two-tier approach of Heron (design the KG in a GUI and implement the Nodes' behaviour in Python) not just a viable but the preferred approach for any experimental designer. Especially for those experimenters who may not be versed in the latest nuances of low-level computer code, but still would like to be fully in control of the behaviour of their experiment. For the cases where a user with deeper knowledge of software engineering has a specific need to use other languages, Heron offers one last benefit arising from the use of its message passing library, 0MQ (*Hintjens, 2013*). 0MQ is a versatile and easy to use library for passing messages across different processes running on different machines. Most importantly, it includes bindings for almost all commonly used languages. Utilising this library (with minimal effort), a user can create an application in any language that communicates with a Heron Node (practically implementing a small part of Heron's protocol in another application). Then a Python wrapper Heron Node can be made responsible for the executable's lifetime. In this way, one has just created, with very little effort, a Node that runs an executable written in some other language but acts just like any other Heron Node passing data to and from any other Nodes.

## Heron's architecture

### Node types

Heron defines three different types of Nodes, each implementing a different basic functionality of message passing. Those are the Sources, the Transforms and the Sinks (see the orange, green, and purple Nodes, respectively, in the example KG of *Figure 1B*). The Sources are Nodes that generate data (either computationally or by reading them from a device) and thus can only transmit data through the Nodes' outputs. The Transform Nodes can both receive and transmit data through both input and output points and are meant to allow data manipulation. Finally, the Sink Nodes can only receive data and only feature input points. The Sink Nodes are designed to either save data or talk to devices that require only computer input and not input from the external world (e.g. a motor). The Nodes' types only exist to generate a cleaner code base by separating the three types of message manipulation (output only, input then output, and input only). There is nothing, though (except Python's rule No 7: Readability counts), stopping a user to create side effects of functions implemented by these Nodes other than message passing, thus interacting with machines in ways different to how the Node's type would suggest.

### Heron's actor-based model

Most of Heron's advantages over similar software tools stem from the way it structures the communication between the different Nodes (and thus processes underlying those). Heron's processes do not allow each other to take control of each other and change each other's state. Each process has full control of its state and will only allow another process to influence it through the passing of messages. This is known as the actor-based model. In contrast, the most commonly used Object-Oriented Programming (OOP) model will allow an object to directly change the state of another. For example, let us consider the situation where the result of an online analysis on incoming data should be used to change a camera's gain. In an OOP world, the object responsible for the analysis would also have to carry a pointer to the memory that represents the object responsible for controlling the camera and directly change the gain (by changing the value of the instance's gain variable) when required. But what happens when another object is introduced later on that also needs to control the gain of the same camera? What if the change of gain is also dependent on the gain's history? Who is responsible for correctly changing this parameter when there is more than one object vying for control, and will the introduction of the new object require changing the

code on both the camera and the analysis object code bases? The actor-based model solves these problems by allowing the camera's gain to be changed only by the object that controls the camera itself. All other objects can only request such a change by sending request messages to the camera object (in Heron's case the camera Node). In this way, when a user composes a Node, they have to think only about what that Node does and how it communicates with other Nodes, and never about the way code outside it might change its behaviour (which Heron with its actor-based model will never allow).

## Heron GUI's multiple uses

To understand Heron's code structure, one must initially appreciate its dual role in designing and running an experiment. When a Graph (short for KG) is not running, Heron acts as a Graph designer, offering a GUI where a user can create and delete Nodes, connect them with Links and assign values to their parameters. During the design period, only one process is active, the one running the Heron GUI (Editor, see *Figure 2*). When, on the other hand, a KG is running, the Heron process stops being a Graph design application and assumes the role of a director in an actor-based model (*Agha, 1986*). It can then concurrently compute and run a GUI for the experiment where the user can update the parameters of the different Nodes on the fly (as an experimental Control Panel). In this actor-based model, each Node is represented by two processes (Worker and Communication, see *Figure 2*), while there are three more processes acting as message forwarders between all other processes (Proof of life, Parameters and Data). That means a running experiment is constituted by (Number of Nodes) × 2 + 4 processes (see *Figure 2*). Each process is an actor that can receive and transmit messages, make local decisions (i.e. decisions that can affect only itself) and determine how to respond to incoming messages. In the (most common) cases where the Nodes running are all implemented by Python code, then the Heron process is responsible for initiating the three forwarders and the com processes for all the Nodes. Each com process will then initiate the corresponding worker process. In the special case (see Rats playing computer games: State machines and non-Python code Nodes example) where a Node will call an executable instead of a Python script, then the worker process can also be responsible for initialising (and terminating) the executable's process. As mentioned above, each Node is represented by two processes. In the code, those are called the com and worker process. The worker process is the one that runs the Node's script defined by the user. The com process is responsible for (1) grabbing messages that come out of other Nodes and are meant to reach the Node (as defined by a Link between two Nodes in the Node Editor), (2) passing those messages to the worker process, (3) receiving any messages the worker process has to pass to other Nodes, and (4) passing those messages to the com processes of all the Nodes that should receive them. The passing of messages between com processes of different Nodes is facilitated by the Data Forwarder process. The worker processes also communicate directly with the Heron process through two separate forwarders. The Parameters forwarder is responsible for passing to the worker processes the parameter values assigned by the user to the processes' respective Nodes on their GUI. This allows the user to also manipulate the state of each Node while an experiment is running. Through this functionality, the Heron GUI becomes (while a Graph is running) also a control centre through which an experimenter can interact with the experiment by changing live the Nodes' parameters. The Proof-of-life forwarder is responsible for receiving an initial message from the worker process which is then passed to the GUI process, letting it know the worker is up and running.

As mentioned above, Heron allows any of the Nodes in a Graph to be initiated and executed in machines different to the one running the main Heron process. At the level of the processes, that means that Heron, if instructed to run a Node on a different machine, will run only the worker process of that Node on the different machine while its com process will run on the same machine as Heron. That has as a drawback that a user cannot put multiple Nodes on a separate machine and expect them to interact (through messages) within that machine since all message passing happens through the com processes which will always run on the Heron running machine. Future versions of Heron will address this limitation by allowing Heron to run Graphs headless (without the GUI process being active), which will allow sub-Graphs to fully run on one machine and communicate their result to Nodes in the machine running Heron.

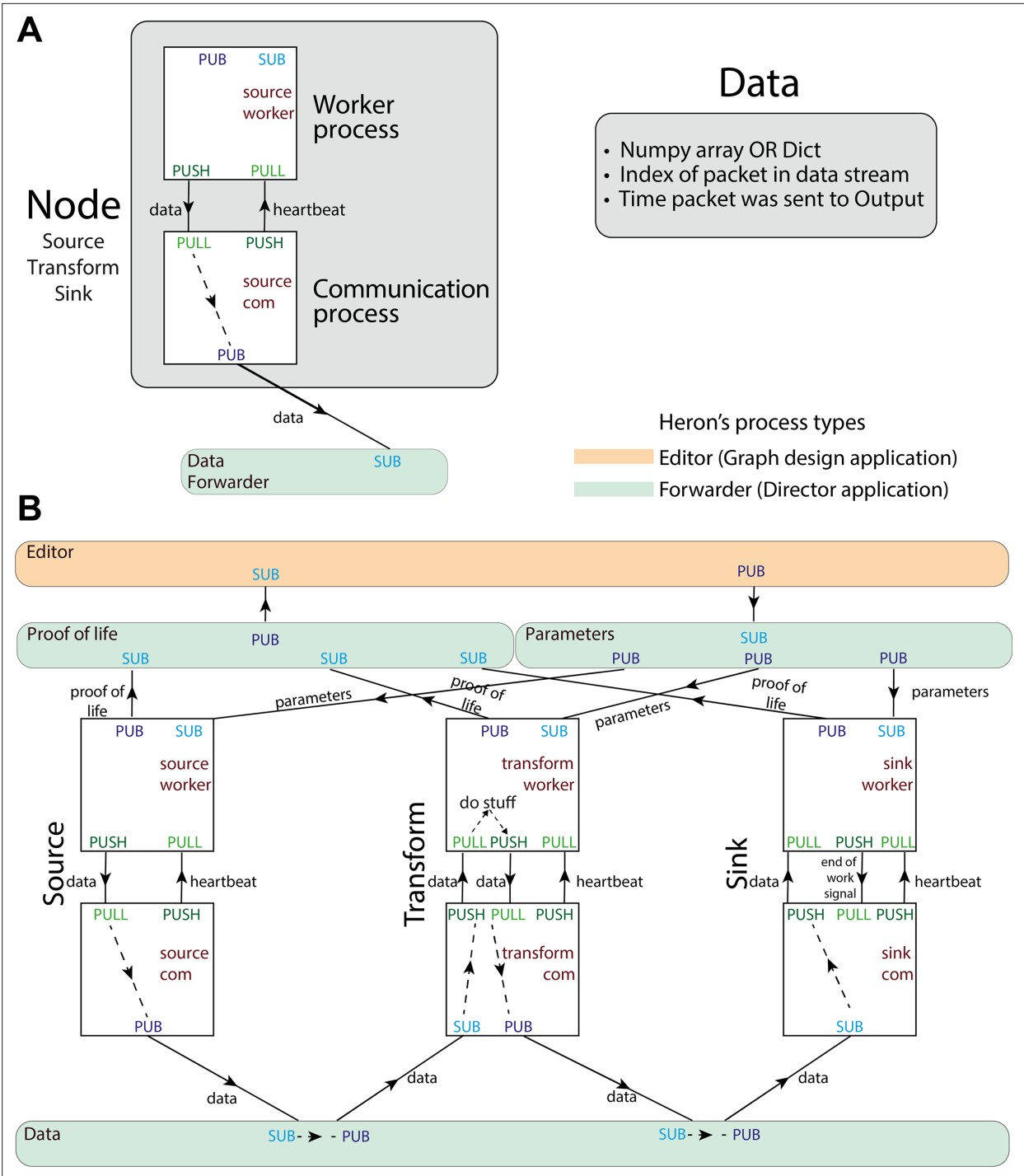

**Figure 2.** Heron's design principles. (**A**) On the left, Heron's Node structure showing the two processes that define a running Node and how messages are passed between them. The design of the special message defined as 'data' is shown on the right. (**B**) Heron's processes and message passing diagram. Each rectangle represents a process. The top orange rectangle is the Heron's Editor process, while the other three green rectangles are the three Forwarders. The Proof-of-life forwarder deals with messages relating to whether a process has initialised correctly. The parameters forwarder deals with messages that pass the different parameter values from the editor process (the Graphical User Interface [GUI]) to the individual Nodes. Finally, the data forwarder deals with the passing of the data messages between the com processes of the Nodes. The squares represent the worker (top) and com (bottom) process of three Nodes (a Source, a Transform, and a Sink from left to right). Solid arrows represent message passing between processes using either the PUB SUB or the PUSH PULL type of sockets as defined in the 0MQ protocol. Dashed line arrows represent data passing within a single process.

## Code architecture

Heron's code is separated into three main folders, each pertaining to one of the three aspects of its basic functionality (see *Appendix 1—figure 1* for all of Heron's folder structures). The Communications folder includes scripts that deal with the low-level communication between all Nodes that make up a running Graph (experiment). The GUI folder holds scripts that deal with Heron's GUI. The Operations folder keeps all the code that defines all the Nodes that Heron recognises and can use to create a Graph with. The Operations folder is further subdivided into the Source, Transform, and Sink folders, each holding codes according to the type of the Node it implements. The Operations folder also holds symbolic links to code repositories separate from Heron. Those, assuming a specific folder structure, are recognised by Heron as valid Node describing codes. The scripts in the Communications folder are class definitions for 8 objects: 6 for the worker and com objects for each Node type which implement Heron's communication protocol, one for the object that deals with the network connectivity (SSHCom) and one re-implementing pyzmq's (Python's 0MQ bindings) Socket object, adding to it the ability to pass numpy arrays and dictionaries as messages. This is required because in Heron all messages are either numpy arrays or Python dictionaries. That means that the worker functions of the worker scripts of the Source and Transform Nodes will always return either a list of numpy arrays or a list of dictionaries. Each element of the list corresponds to one output of the Node. The Operations folder has three levels of subdivision. Immediately below it are the Source, Transform and Sink folders, and inside those are folders representing groups of Nodes in each Node type (e.g. Vision for Nodes that have to do with computer vision). Inside those subcategory folders are the folders that hold the scripts for each Node (e.g. Camera which holds the scripts that read a web camera into Heron). Inside each Node's folder, there are a minimum of two scripts with name suffixes _com and _worker. The _com script allows the user to define a Node's characteristics (parameters, inputs and outputs) with a few lines of code and without requiring any GUI relevant code (Heron takes care of that). The _worker script is responsible for the functionality of the Node (being the script that is run by the Node's worker process through the node_type_worker object) and implements a minimum of three functions. These are the initialisation, the worker and the end-of-life functions (the names are arbitrary and the user can define them as they please). The initialisation function is run when a Node is first started by Heron (i.e. its com process is up and running and its worker process has just started but is being tested before it starts receiving and transmitting data). The worker function is the main function that implements what the Node is supposed to do. The worker functions of the three types of Nodes are implemented differently. In the case of a Source Node, the worker function needs to be an infinite loop that somehow generates data and passes them on to the Node's com process at the end of every loop. The Transform and Sink Nodes need a worker function implemented as a callback since their worker processes will call the worker function every time there is any data arriving at the input of the Nodes (i.e. any time their com process has received a message from another com process and has passed this to its worker process). Both the Transform and the Sink Nodes will stop accepting messages until their worker functions have returned, and Heron is designed to have no message buffering, thus automatically dropping any messages that come into a Node's inputs while the Node's worker function is still running. Finally, the end-of-life function will be called when a worker process hasn't received a heartbeat signal from the source process for a pre-determined amount of time, and its role is to gracefully terminate the process.

## Usage

There are two skills that a user should possess in order to aptly use Heron. Firstly, one requires a familiarity with Heron's GUI which allows (1) downloading and installing new Nodes from existing repositories, (2) defining a local network of computers on which the different Nodes can run, (3) setting up a pipeline using the existing Nodes, and (4) running the pipeline all the while being comfortable in debugging it as problems arise. The second skill is the implementation of new Nodes based on the user's individual needs. In this section, we will provide a basic description of both the GUI usage and the development of new Nodes.

### Using Heron's GUI

### Adding new Nodes from pre-existing code

Heron comes pre-packaged with a small set of Nodes that have a generic enough usage that most users would find useful. An important point though about Heron is that every user will be developing their own Nodes, which in most cases will take the form of code shared in some online repository. Heron is designed to easily access repos that have been developed following a specific file structure to represent a set of Heron Nodes and integrate them into its GUI and workflow without the user needing to do anything else other than create/download the repository and point Heron to it. This also simplifies the further development of Nodes by the community of users since a new Node repository does not have to interact with the main repository of Heron and thus avoids all the pitfalls of pushing, pulling, and merging code repositories at different levels of maturity.

### Local network

Heron's GUI allows an easy definition of the local access network (LAN) of machines that will run Nodes forming a single pipeline. A user has only to provide the IP, port, user name, and password of a machine in the LAN, and Heron will communicate between machines using an SSH protocol, taking care of issues like process lifecycle on different machines, opening and closing ports and proper passing of messaging between processes over the network.

### Setting up a pipeline

Once all the Nodes' repositories have been made known to Heron and the LAN of all machines has been set up, a user needs to implement the experiment's pipeline. This is achieved again graphically by introducing the required Nodes in the Node Editor (main window of Heron), setting up their parameter values and finally connecting the Nodes together by creating links between outputs and inputs. Heron allows many-to-many connectivity, meaning a Node's output can connect to any number of inputs and an input can receive any number of outputs.

### Running a pipeline

Once a pipeline has been defined (generating the KG of the experiment), then running it is achieved by pressing the Start Graph button of Heron. Heron will go through each Node (in order of addition to the Node Editor) and will start the processes that the Node represents (see Heron's Architecture for more details). It will then connect all the processes with 0MQ sockets as defined by the links between the Nodes and pass the Nodes' parameters to the worker process of each Node. The pipeline of data being generated by the Source Nodes, being transformed by the Transform Nodes and finally saved or turned into control of machines by the Sink Nodes will keep on running until the user presses the End Graph button. At this point, Heron will gracefully terminate all processes (including the ones running on separate machines) and close down all communication sockets.

### Creating a new Node

Heron users will develop their own Nodes for their specific experiments. To facilitate this, Heron provides a series of tools. Firstly, there is a set of templates that offer a scaffold on top of which a user can build their own code. The templates have the required code elements that all valid Heron Nodes must possess and are fully documented to help a user quickly build functioning Nodes. An abbreviated and annotated Transform template for the com and worker scripts can be seen in Appendix 2. On top of Node templates, Heron provides a GUI-based system for generating new Nodes. This will create the required files and all the common between Nodes, skeleton code, appropriate to the type of Node under development and using names (for functions and variables) chosen by the user during the GUI-based creation process. Finally, in order to expedite the correct placement of new Nodes in pipelines, Heron offers a Node (User Defined Function 1I 1O) designed to allow Python functions to be run as Nodes in Heron (without the user having to follow Heron's API). This Node can be used to simulate inputs and outputs to an under-development Node in order to test and debug its functionality before incorporating it into a full pipeline.

### Node repositories

As described above, Heron offers the tools to integrate any new code (designed with the correct file structure) into its collection of Nodes and make it available in its GUI. Although not necessary,

good practice would be to develop any new Node (or closely related group of Nodes) as part of a separate repository so that the Node can be easily shared with the rest of the community. Currently, a public GitHub organisation called (rather unimaginatively) Heron-Repositories is hosting both the main Heron Git repository and all other Git repositories of Nodes developed to cover the developers' experimental needs. Any of the individual Heron Nodes repositories can serve as an easy-to-follow example on the file structure expected by Heron for successful integration of new code. All of the Nodes presented in the Results paragraphs examples can be found in this repository.

## Discussion

We have presented Heron, a new tool for coding up experimental pipelines. We have put forward the proposition that using Heron instead of the many other frameworks that one can utilise to create software to run experiments has a series of advantages. It can practically self-document, creating KG that are as close as possible to one's mental schema of the experiment. These KGs and code bases are easy to follow by researchers other than the developers of the experiment, irrespective of the complexity of the experiment. They can trivially connect processes that run on different operating systems and machines in a single, unified pipeline. For example, a series of raspberry pi computers, each reading some cameras or other sensors, can connect and pass the data to a Linux-based, many GPU machine that does online machine learning analysis, while these results can pass to a PC machine running a computer game controlled by those results. It is using a language (Python) for the development of experiments that is one of the easiest and most versatile computer languages with a large community of developers and rich libraries for most functionalities. Finally, it is versatile enough to allow easy integration of code bases written with languages other than the one it has been developed in. We are arguing here that Heron's learning curve, starting from a basic capability in Python, is measured in the few hours of trying to create a couple of new Nodes and joining those together in order to create some toy experiment. Once that is understood, then the limit to what can be achieved is defined by the level of Python knowledge of the developer. Heron has been conceptualised to grow into a community project. Both itself and the repositories holding extra Heron Nodes are open source under an MIT licence. The separation of repositories that hold the main Heron code and the individual Nodes' code bases allows for the growth of a Node ecosystem where users will be able to share their development using standard repository-based tools. Finally, the developers welcome efforts for collaboration with the aim for Heron to eventually become a multi-developer, collaborative project expanding with capabilities covering the needs of experimental scientists in all experimental fields and beyond.

## Materials and methods

Here, we showcase a number of experiments implemented in Heron. Each example has been chosen to highlight one of Heron's competences as described in the Design benefits. As mentioned above, all the Nodes used to construct the examples presented here can be found in the Heron-Repositories GitHub organisation. We are not making public the specific experiment files since these are hardware specific and would need large changes to be made compatible to any other hardware. But Heron's graphical nature makes it easy to go from an image capture of an experiment's Heron GUI (see *Figures 3–5*) to a working experiment by simply combining the required Nodes.

### Probabilistic reversal learning. Implementation as self-documentation

The first experimental example is provided here to showcase how Heron's implementation of an experiment becomes the easiest way for non-developers to acquaint themselves with the experiment and its logic. Thus, here we describe the implementation from the point of view of someone who sees it for the first time and is trying to understand what the experiment is (without accessing any other publication or written explanation). As seen in *Figure 3*, this experimental pipeline is made up of four Nodes. One is called a 'KeyPress', which since it connects to an input named 'Start/Previous Trial Result' seems to play the role of the start button of the experiment. The last Node is a 'Save Pandas DF' which suggests saving the output of the third Node (the 'Trial Controller') in a row of a pandas DataFrame. The main experiment seems to be defined by two Nodes, one named 'Trial Generator' and one named 'Trial Controller'. We can immediately conceptualise the pipeline as a two-part one,

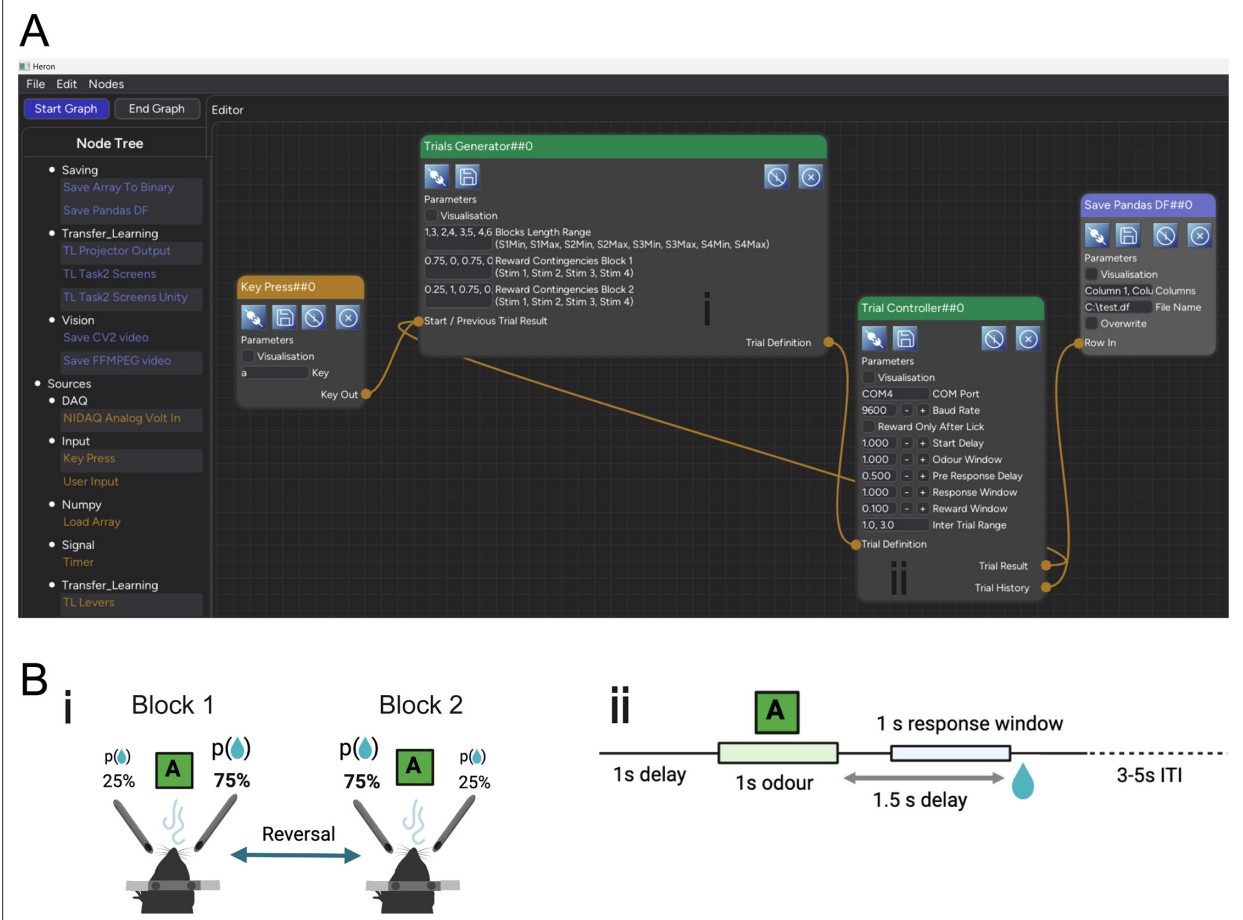

**Figure 3.** The Knowledge Graph of the Probabilistic Reversal Learning experiment. (**A**) The Knowledge Graph of the Probabilistic Reversal Learning experiment showing four Nodes comprising the task, two of which (Trial Generator and Trial Controller) are connected in a loop (the output of one is fed into the input of the other). (**B**) The schematic or mental schema of the experiment. Notice the direct correspondence (**i, ii**) between the Heron Nodes and the two main components of the experiment's mental schema, as well as the Node's parameters and the schema components' basic variables.

The online version of this article includes the following figure supplement(s) for figure 3:

**Figure supplement 1.** The Probabilistic Reversal Learning experiment.

where the first part generates some kind of trial state which it then outputs as its 'Trial Definition' output to the second Node which inputs that 'Trial Definition' and runs (Controls) that trial given its state. We notice that these two Nodes are reciprocally connected, meaning that the 'Trial Generator' requires the output of the 'Trial Controller' (named 'Trial History') to generate the definition of the next trial. Looking a bit more closely at the names of the parameters of the Nodes, we can deduce a number of things about the experiment's structure and function. From the Trial Generator parameters, we see that trials seem to fall into two blocks. We can also see that the experiment has trials with four types of Stim (maybe short for Stimulation, or Stimuli). We can assume (but would need to verify from the code) that the Reward Block Contingencies mentioned for the two Blocks are the probabilities of a reward given the type of Stimulation. So, we have surmised that the trials come into blocks of specific length (probably a random variable drawn from some distribution from the user-provided Blocks Length Range parameter) and of specific trial type (one of four), and each trial type in each block has a user-defined reward probability. So far so intelligible. By looking at the Trial Controller Node's parameters, we see first of all that the Node requires a COM Port and Baud Rate to be defined, showing that it is controlling some device through a serial port. The 'Reward Only After Lick' parameter tells us that this is an experiment where the subject needs to lick (and in some cases this is the only way to get a reward). The names of the rest of the parameters indicate that the experiment is an olfactory one (see 'Odour Window') where the subject gets to experience an odour

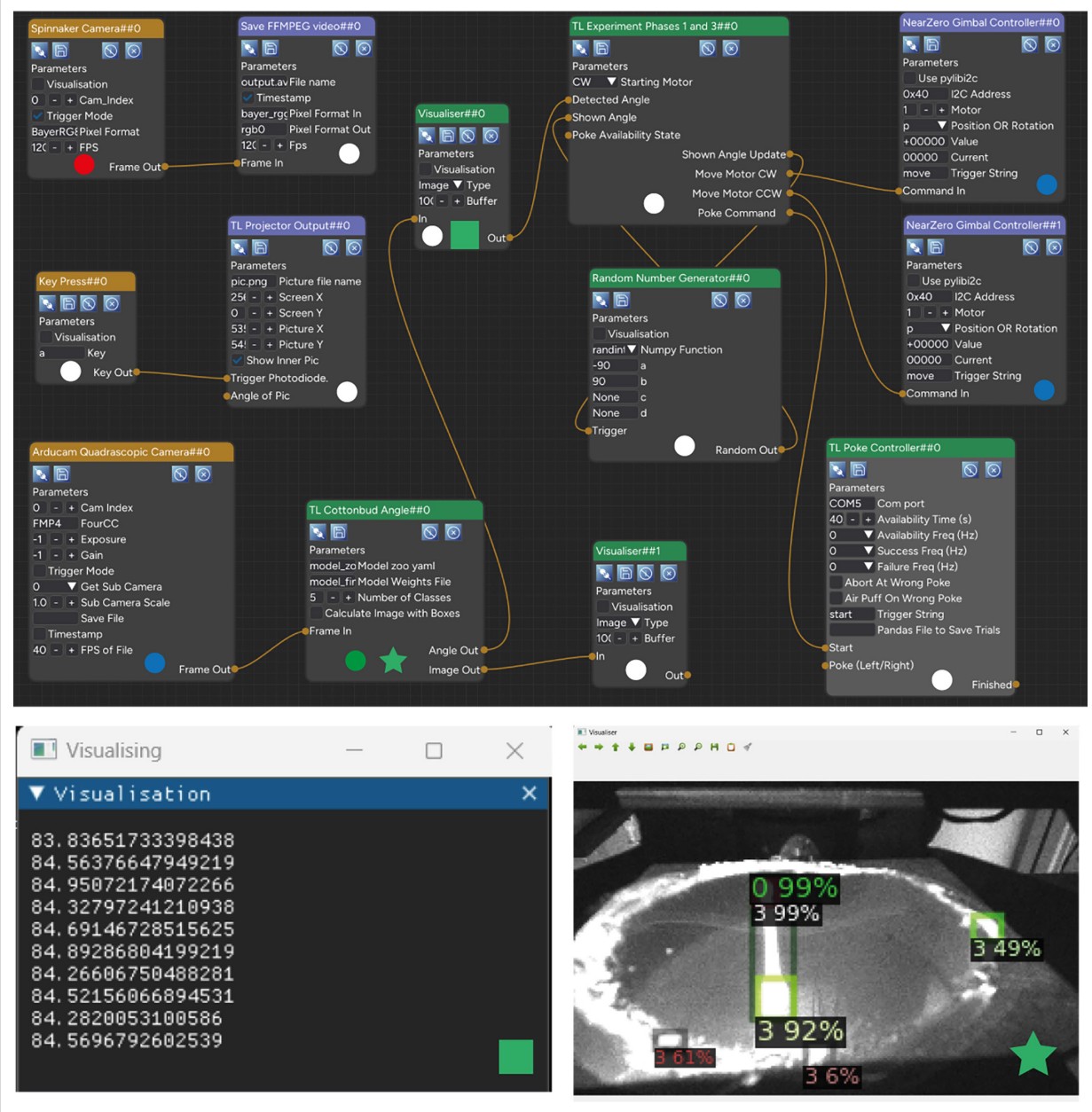

**Figure 4.** The Knowledge Graph of the Fetching the cotton bud experiment. The coloured circles at the bottom middle of each Node and the coloured star and square (where shown) are not part of the Heron Graphical User Interface (GUI) but are used in the figure to indicate in which Machine/OS/ Python configuration each Node is running its worker script (circles) and which visualisation image at the bottom corresponds to which Node. For the circle, the colour code is: White = PC/Windows 11/Python 3.9, Red = PC/Windows 11/Python 3.8, Blue = Nvidia Jetson Nano/Ubuntu Linux/Python 3.9, Green = PC/Ubuntu Linux in WSL/Python 3.9. The two smaller windows below Heron's GUI are visualisations created by the two Visualiser Nodes. The right visualisation (coming from Node Visualiser##1) is the output of the Detectron2 algorithm showing how well it is detecting the whole cotton bud (detection box 0) and the cotton bud's tips (detection boxes 3). The left visualisation box (output of the Visualiser##0 Node) is showing the angle of the cotton bud (in an arbitrary frame of reference where the 0 degrees would be almost horizontal on the screen and the 90 almost vertical). This angle is calculated in the TL Cottonbud Angle Node, which is responsible for running the deep learning algorithm and using its inference results to calculate the angle of the cotton bud. As shown, the TL Cottonbud Angle Node is running on a Linux virtual machine (since Detectron 2 cannot run on Windows machines).

The online version of this article includes the following figure supplement(s) for figure 4:

**Figure supplement 1.** Fetching the cotton bud experiment as controlled by Heron.

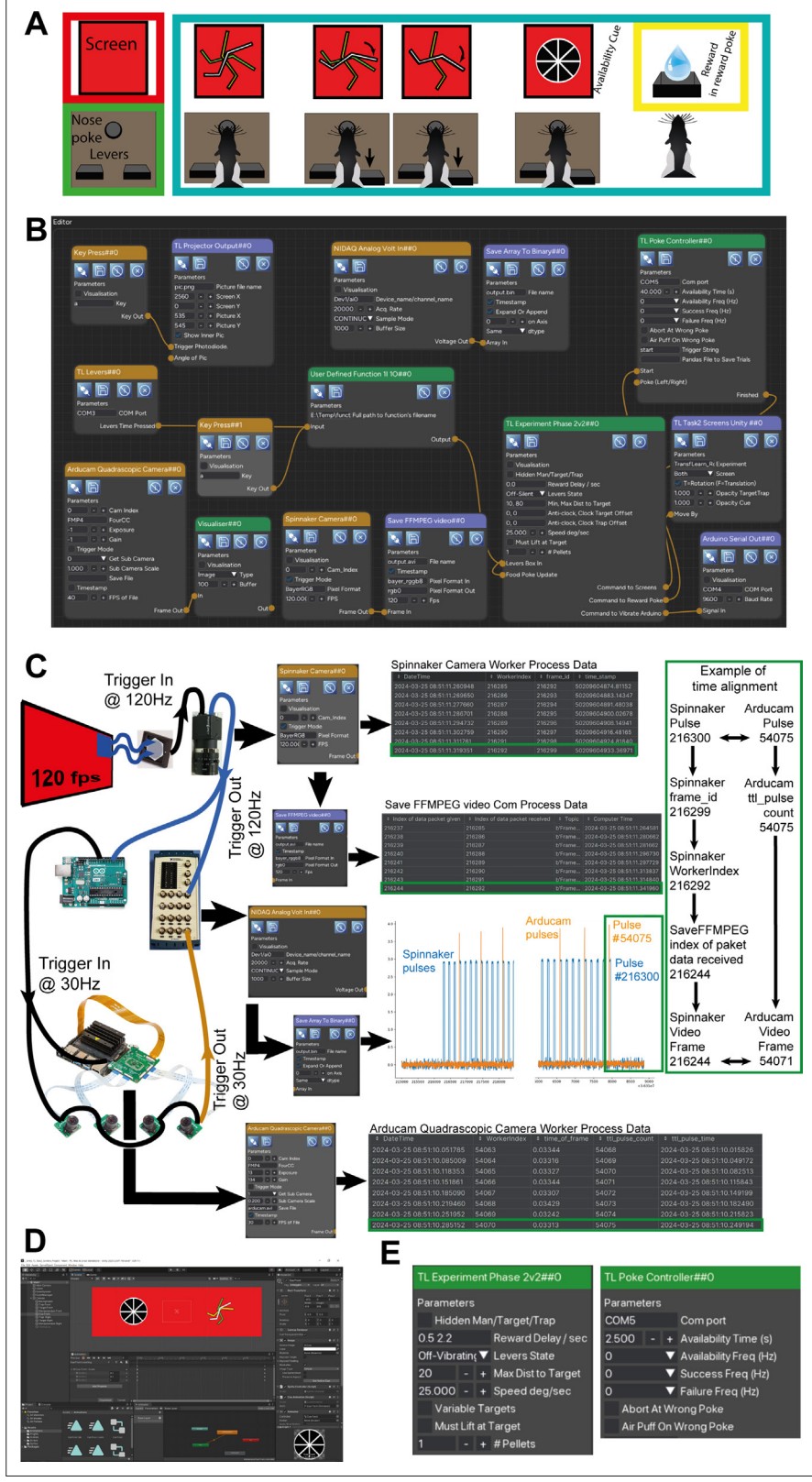

**Figure 5.** Computer game for rats. (**A**) The experimental design (see *Figure 5—figure supplement 1* for a detailed explanation). (**B**) The Heron Knowledge Graph. (**C**) An overview of some of the hardware connectivity of the experimental setup and the way Heron deals with the time alignment of the data packets from different Nodes. The green rectangle to the right shows how one can use Heron's data recording facilities to correlate the frames of

*Figure 5 continued on next page*

*Figure 5 continued*

the Spinnaker Camera saved video (the example here highlights the last frame #216244) with the video frames of the Arducam cameras (#54071 in this example) which are generated and saved by a separate Node running on a different machine. (**D**) The Unity project used to create the visuals and their reactions to the commands sent by the TL Experiment Phase 2v2##0, logic Node. (**E**) A zoom of the parameters of the two logic Nodes in the experiment (TL Experiment Phase2v2##0 and TL Poke Controller##0). Updating those parameters during an experiment, an experimenter can control the way the arena interacts with the subject and allows Heron's Graphical User Interface (GUI) to act as a control panel for the experiment while a Graph is running.

The online version of this article includes the following figure supplement(s) for figure 5:

**Figure supplement 1.** Heron running the 'computer game for rats' experiment.

---

(maybe one of the four stimulations described in the Trial Generator), then gets a pre-response delay, then a response window time and finally a reward window. In order to get a better picture of what the experiment is actually doing, we next look at the code. But most importantly, we know exactly where in the code base we should be looking without needing to search through tens of files to follow an obscure logic to figure out where the bits of the code that implement the actual logic are. There are two Nodes of interest, so we should, as a first step, look at the worker scripts of those two Nodes. The Trial Generator is an undaunting 127 lines of code, while the Trial Controller is 251 (empty lines included). Yet we can zoom even more in where we should initially look by simply ignoring the initialisation and end_of_life functions and just concentrate on the worker_function of each script. Those two functions are shown in the Worker_function Code in Appendix 3. We need only two pieces of prior knowledge to understand the code. One is that the worker functions of the Transform (and Sink) Nodes are callbacks that are called every time the Node receives a message to any of its inputs. Both the Trial Generator and the Trial Controller are Transforms (they have both inputs and outputs and are marked as green on the GUI). The second is that each Node outputs through the return statement of its worker function a list of numpy arrays with each element of the list being the array that is outputted by a specific output of the Node (in order from left to right in the list corresponding from top to bottom in the Node). So, a Node with two outputs will have a worker function that returns a list with two numpy arrays/dictionaries. In this case, for example, the Trial Controller's worker function should return a list with two arrays, i.e. the Trial Result output and the Trial History one. With this in mind and looking at the Trial Generator, we can see that it takes in the previous trial's stimulation number and, if the correct port was licked, uses this information to keep a running track of the correct licks for each port, decide which block we currently are in, decide which stimulation to generate next and offer or not a reward. By following the Trial Controller worker_function's code, we can easily validate our prior speculation about the temporal structure of a single trial following a path of delays interspersed with stimulation and potential reward. An overview of the actual experiment running on Heron, also showcasing how Heron communicates with the Arduino board that does the hardware control, can also be seen in *Figure 3—figure supplement 1*, together with a schematic of the experimental timeline.

## Fetching. Four environments, three operating systems, two machines, one pipeline

This next experiment showcases Heron's ability to run Nodes whose worker process runs on machines separate from the one running the Heron GUI. The experiment itself requires the monitoring of a rodent as it tries to fetch a cotton bud from a random point in a large arena and drop it into a shallow pot under a hole at one of its corners. The hardware setup has a Grasshopper3 USB 3.0 colour camera (GS3-U3-23S6C-C, FLIR) which records the whole of the arena from the top at 120 frames per second with HD resolution, while 4 smaller, black and white, 30 frames per second cameras (OV9281, ArduCam) are used to capture the animal from three different angles and to also monitor the target pot from underneath (since its base is transparent plastic) using the fourth camera. The experiment needs to also know when the animal has deposited the cotton bud and also record at which angle this has happened. When the animal has deposited the cotton bud, it is rewarded with a treat from a nearby reward port. The pot rotates, removing the cotton bud from the arena (while another, empty pot takes its place), and the cotton bud is thrown back into the arena for the animal to fetch again. The Grasshopper camera at the time of the experiment's development could run only on Python 3.8. The four smaller cameras are part of a system that synchronises them, providing a stitched up single

frame of 5120 × 800 pixels resolution, (ArduCam 1MP*4 Quadrascopic Monochrome Camera Kit) which can run only on either a Raspberry Pi (≥3) (Raspberry Pi Foundation), a Jetson Nano (Nvidia) or a Xavier (Nvidia) single board computers. The angle detection of the cotton bud when it is in the pot is done using Meta's Detectron 2 (*Wu et al., 2019*), a deep learning algorithm which we trained with a few hundred samples to detect the cotton bud's edges when in the pot using as an input the part of the OV9281 camera frames that come from the camera underneath the pot. Detectron 2 requires an Intel or AMD CPU-based computer running Linux. Heron itself and all the other Nodes for this experiment run on a Windows PC with Python 3.9. The choice to run the Heron GUI on the Windows machine was made to keep the latency of the 120 fps camera to a minimum. So, the experimental pipeline needs the following Machine/OS/Python configurations: Intel/Windows/Python 3.9+, Intel/Windows/Python 3.8, Intel/Linux/Python 3.9+, and ARM/Linux/Python 3.9+. To create all the above configurations, we first made in the main Windows 11 machine (that Heron runs on in a Python 3.9 environment) a separate (conda) environment with Python 3.8 and the required Spinnaker (for the Flir camera) python package. Then we set up on the same machine a Windows Linux Subsystem (WLS 2) virtual machine, running Ubuntu and Python 3.9 with all the required packages to run Detectron 2. Finally, we connected the Windows machine through an LAN to an NVIDIA Jetson Nano with the Arducam Quadrascopic system running Ubuntu and Python 3.9 with all the packages for the Arducam system to operate. To summarise, the pipeline (which can be seen in *Figure 4*) when run, is utilising two physically separate machines, three operating systems (one Windows 11 and two Linux, one on a virtual machine) and four different Python environments. Once each machine/OS/Python environment is up and running and each one can run the Node(s) that it is supposed to run by itself (something that can be tested and debugged at the level of an individual Node without requiring the whole pipeline to be up and running), then assembling the pipeline is as simple as connecting the Nodes appropriately on Heron's Node Editor and telling each one (through the Node's secondary parameters window). Which computer it should run on and which Python executable it should call to run the worker script. Heron hides from the user all of the work required for the different processes in all the machines to start at the right time, connect correctly to each other, exchange data while the pipeline is running and finally gracefully stop when the pipeline stops without leaving hanging processes and inaccessible bits of memory all over the place. An overview of which machine runs what Node can be seen in *Figure 4* while *Figure 4—figure supplement 1* shows a snapshot of an animal having fetched a cotton bud while its angle is being live detected by one of the Detectron 2 algorithms.

## Rats playing computer games

The rats playing computer games experiment teaches a rat to rotate a line on a screen using a left and a right lever press to hit a target line and avoid a trap line. It presents a rat with a nose poke hole, two levers (left and right to the hole) and two screens to the animal's front and right, when it is poking. At the final stage of training, the animal should be able to first nose poke, look at the screen at a set of jagged lines and press one of the two levers to make one of the lines rotate towards the correct line (target) and away from the wrong one (trap). Once this has been achieved, the jagged lines disappear and a separate visual cue appears (usually animated) letting the animal know that there is reward in the reward port (see *Figure 5—figure supplement 1*). This is a very challenging behaviour for a rat, and the experiment needs several stages of shaping to teach it to the animals. Each stage of shaping towards the final behaviour requires a set of different visual stimuli and a different set of states in a rather large state machine. Also, at the conception of the experiment, there was no prior experience on the ideal path to the final behaviour, so a very malleable stimulus generation technique and state machine development was required in order for a large number of ideas to be tested in a small amount of time. Because of the above, this experiment is ideal in showcasing a series of Heron's capabilities. These are Heron's ability to work with non-Python code, Heron's workload management, the tools Heron provides to keep track of complicated experiments and record all required information and finally Heron GUI's capability to act not just as a Graph designer but also as a control panel while a Graph is running.

## State machines and non-Python code Nodes

As mentioned above, although Heron's code base for Node development is in Python, it is relatively easy to create Nodes with code bases on different languages. One way to do this, (which is the way

most scientific computing Python libraries are using), is to use Python interop technologies which allow functions written in other languages (most commonly languages of the C family) to be called by Python. Although this is very powerful, it is a painstaking task, can be very time consuming, and usually requires significant experience with both Python and whatever other language one is working with. Another, not as a low-level approach, offering less control but faster implementation times, is the use of the 0MQ library to pass messages between a Heron Node and an executable written in another language (what Heron is doing to connect all of the Nodes of a pipeline but with a significantly toned-down communication protocol). Here we took the second route. In the Unity game engine (Unity Technologies, US) we made a simple 2D game using C# code, that covered all the possible visual stimuli we needed to show to the animals at any stage (see *Figure 5D* for a snapshot of the Unity development environment). Instead of using the standard game inputs (e.g. keyboard or game controller) to control what and when was played on the screen, we used a string of commands that was received by the game executable through a 0MQ SUB socket (using NetMQ, one of many C# 0MQ bindings). The Unity-generated executable was also designed to do a handshake through a PUSH PULL socket with whatever process initiated it. This made sure the initialising process knew if anything was wrong and also was able to send some initialisation messages. Once the game executable was ready, we created a standard Heron Sink Node whose initialisation function would start the executable, handshake with it and send it the Node's parameters (e.g. if the game was meant to show its stimuli on one or two screens). Then its worker function would just pass to the correct socket any string that would be received at its input from the other Heron Nodes. Properly formatted strings would be understood by the game and update its state accordingly. The Node's end_of_life function would finally close down the executable when Heron's KG was terminated. This use of process control done not directly from the main Heron process but from a Node's worker process is compatible with the concept of Heron as an actor-based framework where an actor (a Node's worker process) can also initialise and end other actors (in this case the Unity executable). Regarding the state machines used in this experiment, we found that no matter how large and complicated a state machine gets, the difficulty lies in its initial design and not in its implementation. Here, we used statemachine (*Macedo, 2025*), a Python library that allows the definition of a state machine and its individual states with easily attachable callbacks at state transitions.

## CPU load

Using multiple processes rather than multiple threads does incur a hit in the total CPU load Heron will require in order to execute a Graph. The 'Rats playing computer games' experiment, with a 15 node Graph (the largest one we had ever had to execute over multiple experiments implemented in Heron) offers a good test bed to showcase Heron's CPU requirements. The experiment was conducted on a Windows 10 × 64 machine with an 8 core i7-7700K CPU clocking at 4.20 GHz. The Graph runs an acquisition and save of a 120-fps 576 × 676 pixels video, an acquisition and save of a 20-kHz, 5-channel DAQ trace updated at 10 Hz, a Unity game running concurrently on two HD screens updating at 120 Hz and an acquisition and save of a 30-fps 5460 × 800 pixels video running on the Jetson Nano machine. When the computer was idle, it showed a base CPU load of just around 10–15%. When Heron was up but not running a KG, it consumed about 2–5% of the CPU for a total load of 15–20%. With the above experiment running, the CPU load hit 100% with the Unity game requiring around 24% and the remaining 45–55% used by the Graph's 32 processes (one process was running on an NVIDIA Jetson Nano machine). The above numbers, though, are true if the operating system was allowed to assign CPU cores to the different Heron operations as it saw fit. Heron has a feature which allows the user to force the operating system to keep a worker process running on a specific CPU core. Rerunning the above Graph while setting 10 of the 15 worker processes to specific CPUs and allowing the system to optimise the load for only 5 of them (the Spinnaker Camera, Save FFMPEG Video, Arducam Quadrascopic Camera, Save Array To Binary, and TL Taks2 Screens Unity) dropped the total CPU load to 80%, again with the Unity game consuming around 24% while all Graph python processes consuming around 25%. The above specifications show that Heron is capable of running Graphs with heavy acquisition and save loads. Also, Heron's ability to utilise multiple machines in a single Graph means that processes requiring exceptional CPU, memory and/or GPU loads can be offloaded to machines other than the one running the Heron GUI. In the above example, this is the case with the acquisition and save of the Arducam video. Also, before implementing the CPU setting

feature, we used to run the Unity game on a 3rd machine in order to keep the 24% CPU load requirement out of the main Heron machine. We stopped using this 3rd machine once the CPU setting feature was in place, and the transition from three to two machines required only a single line change in the Node's secondary window to indicate where the worker script was to be found.

## Synchronisation (time alignment)

Multiple processes, running over different machines, can become extremely difficult to synchronise, especially to the tight requirements of a scientific experiment. Heron offers a set of tools to allow users to time-align every single packet of data generated and saved by any Node to any other packet irrespective of whether the packets were generated by Nodes on the same or separate machines. Here we use the 'Rats playing computer games' experiment to showcase how such time alignment would be achieved between a few of the experiment's Nodes.

More specifically, the experiment has two separate video captures which, since they happen on different machines, can only be synchronised through an external clock. This is similar to an electrophysiology data acquisition system whose data points can be synchronised to other data sets (e.g videos, or button presses), only by an external clock (a TTL pulse generator usually) that is simultaneously recorded by all devices. The whole setup is shown diagrammatically in *Figure 5C*.

We use a 120-fps projector (whose output is controlled by the Node TL Projector Output##0) to act as our base clock. By pressing a key after the Graph has started running, a part of the projector's output (depicted as a screen in *Figure 5C*) turns blue and thus stimulates a blue sensitive photodiode. This happens at every frame, thus the photodiode generates a 120-Hz TTL pulse train timed to the onset of the blue pass of the projector. That train starts when the key is pressed and stops when it is pressed a second time (thus demarcating the start and stop of the recording part of the session). The photodiode's TTL pulse acts as an external trigger to the Spinnaker (FLIR) camera whose output is being captured by Heron's Spinnaker Camera##0 Node. The same pulse is also fed to a National Instruments USB Data Acquisition System (NIDAQ) and to an Arduino. The NIDAQ is being captured (at 10 packets of 2000 points each per second) by Heron's NIDAQ Analog Volt In##0 Node. The Arduino runs a small loop that generates a second TTL pulse for every four TTL pulses coming in from the Spinnaker camera (thus creating a 30-Hz TTL train). This TTL train is fed as an external trigger to the four cameras of the Arducam system as well as to the GPIO of the Jetson Nano machine that is running the Heron Node which captures the Arducam's video frames output (Arducam Quadrascopic Camera ##0). The Arducam system generates a TTL pulse every time the global shutters of the four cameras fire, and that is fed into the same NIDAQ as the Spinnaker TTL pulse out (see left column of *Figure 5C*).

The Spinnaker camera at every trigger generates not only a frame but also the frame's id. This is captured by Heron's Spinnaker Node's Worker process and saved to its Substate dataframe together with the equivalent id of the Node's data packet and the timestamp of this (see *Figure 5C* 'Spinnaker Camera Worker Process Data' table). A Substate dataframe is Heron's way to allow the designer of a Node to save information (to a pandas dataframe) every time the Worker process runs a pass through its loop. This is done through Heron's API, requiring only a single line of code with the information to be saved to the dataframe.

Any failures of the Node to capture a frame get registered in this correlation between camera ids and packet ids. The Spinnaker Node sends the captured frame to the SaveFFMPEGvideo##0 where an ffmpeg pipeline saves each frame to a video stream. As mentioned before, Heron does not offer any capability to buffer packets, so if the FFMPEG Node is too late to save any frame, then the next one gets lost. Heron records this in the FFMPEG Node's Com process log file. A Com processes log file (optionally set for any Node by filling in the log file's path to the Node's secondary window appropriate entry) records the id of any packet that arrives from an upstream Node as given by that Node and the id of that packet as given by the current Node. In this way, any dropped packets are registered, and a one-to-one equivalence between packets sent and packets received can always be established. In this case, the FFMPEG Node's Com process log file indicates the id of each packet that has left the Spinnaker Node and arrived in the FFMPEG Node and the id assigned to it by the FFMPEG Node. The id of the FFMPEG Node corresponds to the frame id of the video generated by the ffmpeg pipeline in the Node. In this way, we can now correlate any frame of the produced video to the id of that frame generated by the Spinnaker camera, despite some dropped frames in the process.

In the shown example, the camera generated a total of 216300 frames, and the final video had a total of 216,244 frames, a drop of 56 frames dropped over the two Nodes, that is a 0.02%.

At the same time, the Arducam system (triggered by the sub-sampled TTL pulse from the Spinnaker camera) generates frames at a rate of 30 fps. These are captured and saved by the Arducam Node running on the Jetson Nano. The Node is also registering (for every incoming frame) the equivalent id of the TTL pulse that generated that frame by also reading the Jetson Nano's GPIO port. In this way, the Arducam Node's Worker process can generate a Substate dataframe that correlates the packet's id as given by the process (and thus the id of the video's frame) with the TTL pulse id that produced that frame.

In order to now correlate the frames of the Arducam video with those of the Spinnaker video, we use the saved TTL pulses from the NIDAQ instrument as saved by the Heron NIDAQ Node. The id of the frame captured by the Spinnaker camera corresponds to the number of the pulse in the Spinnaker pulse train, and the id of the pulse saved in the Jetson Nano's GPIO port corresponds to the number of the pulse in the Arducam pulse train. By knowing which Spinnaker pulse has generated which Arducam pulse, one can work backwards through the above described correlations all the way to knowing which frame in the Spinnaker video corresponds to which frame in the Arducam one, despite those being captured on different devices and some frames getting lost by Heron's 'dropping of packets' architecture. In *Figure 5C*, we show more specifically how the last pulses of the TTL trains in the NIDAQ can be used to trace the correspondence between the last frames in the videos of the Spinnaker and Arducam systems using Heron's Substate dataframes and log files.

## On-the-fly experimental control

As mentioned above, the 'Rats playing computer games' experiment is a rather demanding task that requires slow and careful shaping on the subjects' behaviour. Certain parts of the training could not be automated (especially when we were trying ideas that we did not know if they were relevant to the final shaping protocol). During those sessions, the experimenter had to change a number of the experiment's parameters on the fly as the subjects produced behaviour, given both how well they were doing and how the experimenter perceived certain aspects of their behaviour that were too hard to quantify (and thus use in an automated system). Examples of these experimental parameters were the number of reward pellets offered to the subjects, the maximum angle between the moving line and the target line and the speed of the moving line (and thus the amount of time required for a lever to be kept pressed), the amount of time the reward was available after a successful trial and whether the levers would vibrate or not when pressed correctly, etc. (see *Figure 5E*). Heron allowed us to operate in this 'experimenter in the loop' fashion because it allows the Worker process of any Node to update its parameters as they are changed by the experimenter through the GUI while the Graph is running. Which parameters are capable of online updating is up to the author of the Node, and Heron allows both updatable and non-updatable parameters (which are easily defined in the Worker process code through Heron's API).

## Documentation and repositories

The code for all repositories used in this report can be found at the Heron GitHub (*Dimitriadis, 2025*), while Heron's documentation is hosted at Herton docs. Heron's name is a tribute to one of the first known creators of automata, Hero of Alexandria.

## Ethical approval and animal details

### Ethical approval

All experiments were performed in accordance with the UK Home Office regulations Animal (Scientific Procedures) Act 1986 and the Animal Welfare and Ethical Review Body (AWERB).

## Animal details

For the Fetching and Computer Games experiments, subjects were adult male Lister Hooded rats (Charles River), which were kept on a reversed 12-hr light–dark cycle. Rats were food-restricted and received chocolate pellet reward while performing the behavioural task. A total of 10 were used in the behavioural experiments. Mouse experiments for the Probabilistic Reversal Learning experiment were conducted using male WT C57/BL6 (Charles River), aged at 6–8 weeks at the start of experiments.

Mice were housed in cages of two to four animals under a 12-hr light/dark cycle. Mice had ad libitum access to food, but were water restricted to 85% of base weight during behavioural training and received water rewards in the experiment. A total of four mice were used.

## Additional information

### Competing interests

Athena Akrami: Reviewing editor, *eLife*. The other authors declare that no competing interests exist.

### Funding

| Funder | Grant reference number | Author |
|---|---|---|
| Gatsby Charitable Foundation | GAT3755 | George Dimitriadis Athena Akrami |
| Wellcome Trust | 219627/Z/19/Z | George Dimitriadis Athena Akrami |

The funders had no role in study design, data collection, and interpretation, or the decision to submit the work for publication. For the purpose of Open Access, the authors have applied a CC BY public copyright license to any Author Accepted Manuscript version arising from this submission.

### Author contributions

George Dimitriadis, Conceptualization, Software, Methodology, Writing – original draft, Writing – review and editing, Project administration, Validation; Ella Svahn, Validation; Andrew F MacAskill, Resources; Athena Akrami, Resources, Supervision

### Author ORCIDs

George Dimitriadis ⓘ https://orcid.org/0000-0002-1419-1173
Ella Svahn ⓘ http://orcid.org/0000-0003-1976-1531
Andrew F MacAskill ⓘ https://orcid.org/0000-0002-0196-3779
Athena Akrami ⓘ https://orcid.org/0000-0001-5711-0903

### Ethics

All experiments were performed in accordance with the UK Home Office regulations Animal (Scientific Procedures) Act 1986 and the Animal Welfare and Ethical Review Body (AWERB). For the Fetching and Computer Games experiments subjects were adult male Lister Hooded rats, which were kept on a reversed 12-hr light–dark cycle. Rats were food-restricted and received chocolate pellet reward while performing the behavioural task. A total of 10 were used in the behavioural experiments. Mouse experiments for Probabilistic Reversal Learning experiment were conducted using male WT C57/BL6 (Charles River), aged at 6–8 weeks at the start of experiments. Mice were housed in cages of two to four animals under a 12-hr light/dark cycle. Mice had ad libitum access to food, but were water restricted to 85% of base weight during behavioural training and received water rewards in the experiment. A total of four mice were used.

Reviewer #2 (Public review): https://doi.org/10.7554/eLife.91915.3.sa1
Author response https://doi.org/10.7554/eLife.91915.3.sa2

## Additional files

### Supplementary files

MDAR checklist

### Data availability

No data were generated. All code is open source and available on GitHub (https://github.com/Heron-Repositories).

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

## Appendix 1

## Heron folder structures

**A**

```
Heron
    Heron
        Operations
        communication
            forwarders.py
            sink_com.py
            sink_worker.py
            socket_for_serialization.py
            source_com.py
            source_worker.py
            ssh_com.py
            ssh_info.json
            transform_com.py
            transform_worker.py
        docs
        gui
            create_new_node.py
            editor.py
            fdialog.py
            message.py
            node.py
            operations_list.py
            save_node_state.py
            settings.py
            ssh_info_editor.py
            visualisation_dpg.py

        resources
        templates
        settings_default.json
        constants.py
        general_utils.py
```

**C**

```
My_awesome_Nodes_repo
    README.md
    Sources
        Vision
            __top__
                ignore.gitignore
        Weird_Camera
            weird_camera_com.py
            weird_camera_worker.com
    Transforms
        Motion
            __top__
                ignore.gitignore
        Super_Motor_Controller
            Maybe_Another_Folder
                another_script.py
            super_motor_controller_com.py
            super_motro_controller_worker.py
    Sinks
        Saving
            __top__
                ignore.gitignore
        Saving_CSVs
            some_other_script.py
            saving_csvs_com.py
            saving_csvs_worker.py
        General
            __top__
                ignore.gitignore
        Some_Other_Sink_Node
            some_other_sink_node_com.py
            some_other_sink_node_worker.py
```

**B**

```
Base_repository_folder
    Sources / Transforms / Sinks
        Subcategory
            __top__
                ignore.gitignore
        Name_of_Node
            com_script.py
            worker_script.py
```

**Appendix 1—figure 1.** Heron's folder structures. (**A**) The Heron folder structure together with the basic scripts that comprise Heron's core functionality. (**B**) The Heron Node repository folder structure (to be found in the Operations folder for each group of Nodes in the same repository). (**C**) An example of a repository with four Nodes: the Weird_Camera Source in subcategory Vision, the Super_Motion_Controller Transform in subcategory Motion and the Sinks Saving_CSVs in Saving and Some_Other_Sink_Node in General. The __top__ folder with the file ignore.ignore git file is required to allow Heron to create the list of Nodes presented to the users live as its folder structure is updated with new Node folders.

## Appendix 2

### Com template

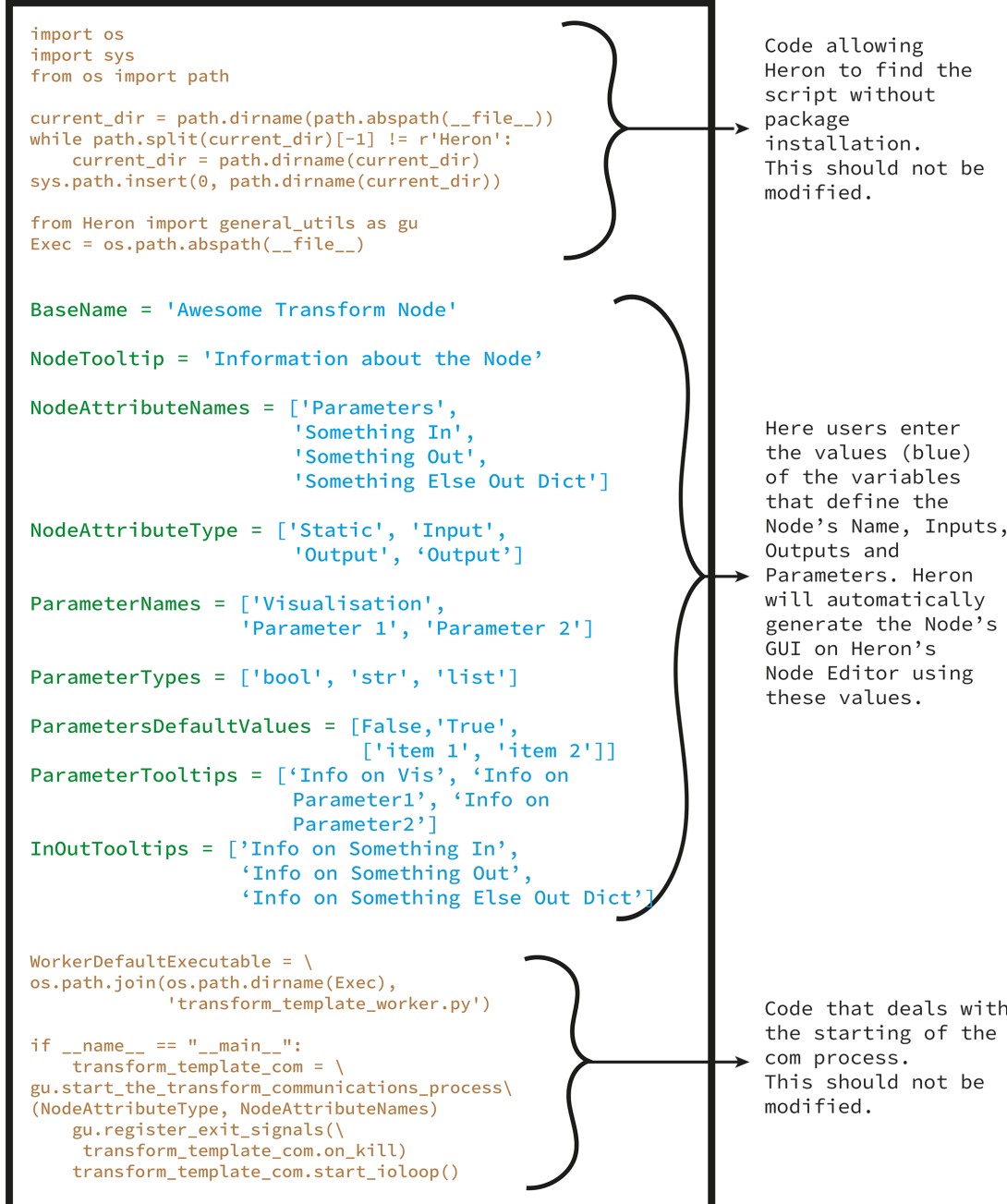

**Appendix 2—figure 1.** An abbreviated and annotated template for the com script of a Transport Operation.

```python
import sys
from os import path
import numpy as np

current_dir = path.dirname(path.abspath(__file__))
while path.split(current_dir)[-1] != r'Heron':
    current_dir = path.dirname(current_dir)
sys.path.insert(0, path.dirname(current_dir))

from Heron.communication.socket_for_serialization import Socket
from Heron import general_utils as gu, constants as ct
from Heron.gui.visualisation import VisualisationDPG

need_parameters = True
global_var_1: str
global_var_2: str

vis: VisualisationDPG

def initialise(worker_object):
    global vis
    global need_parameters
    global global_var_1
    global global_var_2

    try:
        parameters = worker_object.parameters
        global_var_1 = parameters[1]
        global_var_2 = parameters[2]
    except:
        return False

    visualisation_type = 'Value'
    buffer = 20
    vis = VisualisationDPG(_node_name=worker_object.node_name,
                           _node_index=worker_object.node_index,
                           _visualisation_type=visualisation_type,
                           _buffer=buffer)

     worker_object.savenodestate_create_parameters_df(
            parameter_var_1=global_var_1,
            parameter_var_2=global_var_2,
            parameter_var_3=global_var_3,
            parameter_var_4=global_var_4)

    return True

def work_function(data, parameters):
    global vis
    global global_var_1
    global global_var_2

    try:
        global_var_1 = parameters[1]

        vis.visualisation_on = parameters[0]
    except:
        pass

    topic = data[0]
    print(topic)

    message = data[1:]
    data = Socket.reconstruct_array_from_bytes_message(message)

    # Now do stuff

    savenodestate_update_substate_df(image__shape=message.shape)

    vis.visualise(message)

    result = [message, np.array([ct.IGNORE])]

    return result

def on_end_of_life():
    global vis

    vis.kill()

if __name__ == "__main__":
    worker_object = \
gu.start_the_transform_worker_process(\
    work_function=work_function,
    end_of_life_function=on_end_of_life,
    initialisation_function=initialise)

    worker_object.start_ioloop()
```

Code allowing
Heron to find the
script without package
installation.
This should not be modified.

Global variables is just
one of the possible ways
for the intialise,
work_function and
on_end_of_life functions
to keep state over
multiple calls.

The initialise
function is called at the
start of the process
recurently until it returns
True.
The worker_function
will not be called
until the initialise
function returns True.
Here goes the code that
deals  with the
initialisation of the
worker process (like
grabbing the values of the
parameters from the Node's
GUI).

The worker_function is
where the main code of
the Operation should be
placed. It will be called
by Heron every time there
is data coming into the
Node.
It will return a list of
numpy arrays. The items of
that list is what the Node
sends from its outputs
(each item corresponding to
one output).

The end_of_life function
will be called by Heron
when the process is about
to terminate so any code
to clean up goes here.

This is the code that is
called by the com process
to start the worker process.
This should not be modified.

**Appendix 2—figure 2.** An abbreviated and annotated template for the worker script of a Transport Operation.

## Appendix 3

### Com code example

```
def work_function(data, parameters):
    global vis
    global lengths_blockx
    global current_block
    global reward_contingencies
    global correct_licks

    vis = parameters [0]

    message = data[1:] # data[0] is the topic
    message =
    Socket.reconstruct_array_from_bytes_message_cv2correction(message)

    # If the message comes from the Trial controller
    if len (message)==2:
        # The Trial controller sends [previous_stim, correct_port_licks]
        previous_stim = message[0]
        correct_port_licks = message [1]

        # Add the number of correct licks to the current running sum
          correct_licks[previous_stim]=
          correct_licks[previous_stim] + correct_port_licks

        # If the number of correct licks reaches the block length of
        # that stim then swap block and zero the running sum of
        # licks for that stim
        if correct_licks[previous_stim]==
          lengths_block[current_block[previous_stim]][previous_stim]:
            create_new_block_sizes(previous_stim)
            temp = copy.copy(current_block[previous_stim])
            current_block[previous_stim]=
            int (not current_block[previous_stim])
            correct_licks[previous_stim]=0
      if vis:
            print(f'Changing block of stim {previous_stim},
              from block {temp} to
              block {current_block[previous_stim]}')
            print(f'Current block lengths = {lengths_block}')

    current_stim = np.random.randint(0, 4)
    current_correct_reward_port_probability =
          reward_contingencies[current_block[current_stim]][current_stim]
    current_correct_reward_port =
```

```
np.random.binomial(1,current_correct_reward_port_probability)

        result = [np.array([current_stim,
                current_correct_reward_port,
                current_block[current_stim]])]
        return result
```

## Worker code example

```python
def work_function(data, parameters):
global vis
global respond_after_lick
global start_delay
global odour_window
global pre_response_delay
global response_window
global reward_window
global inter_trial_window
global trial_number
trial_number +=1

try:
        vis = parameters[0]
except:
        pass

# Get message in from previous Node
message = data[1:] # data[0] is the topic
message =
Socket.reconstruct_array_from_bytes_message_cv2correction(message)
stim = message[0]
reward_port = message [1]
block_of_stim = message [2]

if vis:
        print (f'==========Starting Trial {trial_number} =========')
        print(f'Current Stim = {stim},
        current correct reward port = {reward_port}')

start_trial_time = now()
if vis:
      print (f' ooo Waiting Start Delay {start_delay}')
 gu.accurate_delay(start_delay * 1000)

open_stim_time = now()

if vis:
      print('====Arduino Open Stim {}'.format (stim))
 arduino_serial.write(stim_on_commands[stim])

if vis:
      print (f'ooo Starting Odour Delay {odour_window}')
gu.accurate_delay(odour_window * 1000)
```

```
close_stim_time = now()
if vis:
 print (f'===Arduino Close Stim {stim}')
arduino_serial.write(stim_off_commands[stim])

if vis:
 print (f'ooo Starting Pre-Response Delay{pre_response_delay}')
gu.accurate_delay(pre_response_delay * 1000)

correct_port_lick, start_response_time = read_arduino(reward_port)
if respond_after_lick and correct_port_lick:
    start_reward_time, end_reward_time =
    reward(correct_port_lick, reward_port)
elif respond_after_lick and not correct_port_lick:
    if vis:
        print(" ooo Correct port hasn't been licked = No reward")
    start_reward_time = datetime.datetime.now()
    end_reward_time = start_reward_time
elif not respond_after_lick:
    start_reward_time, end_reward_time =
    reward(correct_port_lick, reward_port)

if vis:
    print (f'ooo Starting Response Delay {reward_window}')
gu.accurate_delay(reward_window * 1000)
iti = np.random.uniform(inter_trial_window[0], inter_trial_window[1])

if vis:
    print (f'ooo Starting ITI Delay {iti}')
gu.accurate_delay(iti * 1000)
end_iti_time = now()
result = [np.array([stim, correct_port_lick]),
            np.array([start_trial_time, open_stim_time,
            close_stim_time,
            start_response_time, start_reward_time,
            end_reward_time, end_iti_time, str (stim),
            str (reward_port), str (block_of_stim),
            str (correct_port_lick)])]

if vis:
    print (f' ooo Stim and Correct Port Lick = {result[0]}')
    print ('============Ended Trial{trial_number}==============')
    print ('————————————–————————————–————————')
return result
```

