## [Editor Report · eLife Assessment]

This **valuable** paper introduces Heron, lightweight scientific software that is designed to streamline the implementation of complex experimental pipelines. The software is tailored for workflows that require coordinating many logical steps across interconnected hardware components with heterogeneous computing environments. The authors **convincingly** demonstrate Heron's utility and effectiveness in the context of behavioral experiments, addressing a growing need among experimentalists for flexible and scalable solutions that accommodate diverse and evolving hardware requirements.

---

## [Referee Report · Reviewer #2 (Public review)]

Summary:

The authors provide an open-source graphic user interface (GUI) called Heron, implemented in Python, that is designed to help experimentalists to:

(1) Design experimental pipelines and implement them in a way that is closely aligned with their mental schemata of the experiments

(2) Execute and control the experimental pipelines with numerous interconnected hardware and software on a network.

The former is achieved by representing an experimental pipeline using a Knowledge Graph and visually representing this graph in the GUI. The latter is accomplished by using an actor model to govern the interaction among interconnected nodes through messaging, implemented using ZeroMQ. The nodes themselves execute user-supplied code in, but not limited to, Python.

Using three showcases of behavioral experiments on rats, the authors highlighted four benefits of their software design:

(1) The knowledge graph serves as a self-documentation of the logic of the experiment, enhancing the readability and reproducibility of the experiment,

(2) The experiment can be executed in a distributed fashion across multiple machines that each has different operating system or computing environment, such that the experiment can take advantage of hardware that sometimes can only work on a specific computer/OS, a commonly seen issue nowadays,

(3) The users supply their own Python code for node execution that is supposed to be more friendly to those who do not have a strong programming background,

(4) The GUI can also be used as an experiment control panel for users to control/update parameters on the fly.

Strengths:

(1) The software is light-weight and open-source, provides a clean and easy-to-use GUI,

(2) The software answers the need of experimentalists, particularly in the field of behavioral science, to deal with the diversity of hardware that becomes restricted to run on dedicated systems. It can also be widely adopted in many other experimental settings.

(3) The software has a solid design that seems to be functionally reliable and useful under many conditions, demonstrated by a number of sophisticated experimental setups.

(4) The software is well documented. The authors pay special attention to documenting the usage of the software and setting up experiments using this software.

Comments on revisions: The authors have addressed my concerns from the initial review.

---

## [Author Response]

The following is the authors’ response to the original reviews.

**Public Reviews**

**Reviewer #1 (Public Review):**
Summary:The authors have created a system for designing and running experimental pipelines to control and coordinate different programs and devices during an experiment, called Heron. Heron is based around a graphical tool for creating a Knowledge Graph made up of nodes connected by edges, with each node representing a separate Python script, and each edge being a communication pathway connecting a specific output from one node to an iput on another. Each node also has parameters that can be set by the user during setup and runtime, and all of this behavior is concisely specified in the code that defines each node. This tool tries to marry the ease of use, clarity, and selfdocumentation of a purely graphical system like Bonsai with the flexibility and power of a purely code-based system like Robot Operating System (ROS).Strengths:The underlying idea behind Heron, of combining a graphical design and execution tool with nodes that are made as straightforward Python scripts seems like a great way to get the relative strengths of each approach. The graphical design side is clear, selfexplanatory, and self-documenting, as described in the paper. The underlying code for each node tends to also be relatively simple and straightforward, with a lot of the complex communication architecture successfully abstracted away from the user. This makes it easy to develop new nodes, without needing to understand the underlying communications between them. The authors also provide useful and well-documented templates for each type of node to further facilitate this process. Overall this seems like it could be a great tool for designing and running a wide variety of experiments, without requiring too much advanced technical knowledge from the users.The system was relatively easy to download and get running, following the directions and already has a significant amount of documentation available to explain how to use it and expand its capabilities. Heron has also been built from the ground up to easily incorporate nodes stored in separate Git repositories and to thus become a large community-driven platform, with different nodes written and shared by different groups. This gives Heron a wide scope for future utility and usefulness, as more groups use it, write new nodes, and share them with the community. With any system of this sort, the overall strength of the system is thus somewhat dependent on how widely it is used and contributed to, but the authors did a good job of making this easy and accessible for people who are interested. I could certainly see Heron growing into a versatile and popular system for designing and running many types of experiments.Weaknesses:(1) The number one thing that was missing from the paper was any kind of quantification of the performance of Heron in different circumstances. Several useful and illustrative examples were discussed in depth to show the strengths and flexibility of Heron, but there was no discussion or quantification of performance, timing, or latency for any of these examples. These seem like very important metrics to measure and discuss when creating a new experimental system.

Heron is practically a thin layer of obfuscation of signal passing across processes. Given its design approach it is up to the code of each Node to deal with issues of timing, synching and latency and thus up to each user to make sure the Nodes they author fulfil their experimental requirements. Having said that, Heron provides a large number of tools to allow users to optimise the generated Knowledge Graphs for their use cases. To showcase these tools, we have expanded on the third experimental example in the paper with three extra sections, two of which relate to Heron’s performance and synching capabilities. One is focusing on Heron’s CPU load requirements (and existing Heron tools to keep those at acceptable limits) and another focusing on post experiment synchronisation of all the different data sets a multi Node experiment generates.

(2) After downloading and running Heron with some basic test Nodes, I noticed that many of the nodes were each using a full CPU core on their own. Given that this basic test experiment was just waiting for a keypress, triggering a random number generator, and displaying the result, I was quite surprised to see over 50% of my 8-core CPU fully utilized. I don’t think that Heron needs to be perfectly efficient to accomplish its intended purpose, but I do think that some level of efficiency is required. Some optimization of the codebase should be done so that basic tests like this can run with minimal CPU utilization. This would then inspire confidence that Heron could deal with a real experiment that was significantly more complex without running out of CPU power and thus slowing down.

The original Heron allowed the OS to choose how to manage resources over the required process. We were aware that this could lead to significant use of CPU time, as well as occasionally significant drop of packets (which was dependent on the OS and its configuration). This drop happened mainly when the Node was running a secondary process (like in the Unity game process in the 3rd example). To mitigate these problems, we have now implemented a feature allowing the user to choose the CPU that each Node’s worker function runs on as well as any extra processes the worker process initialises. This is accessible from the Saving secondary window of the node. This stops the OS from swapping processes between CPUs and eliminates the dropping of packages due to the OS behaviour. It also significantly reduces the utilised CPU time. To showcase this, we initially run the simple example mentioned by the reviewer. The computer running only background services was using 8% of CPU (8 cores). With Heron GUI running but with no active Graph, the CPU usage went to 15%. With the Graph running and Heron’s processes running on OS attributed CPU cores, the total CPU was at 65% (so very close to the reviewer’s 50%). By choosing a different CPU core for each of the three worker processes the CPU went down to 47% and finally when all processes were forced to run on the same CPU core the CPU load dropped to 30%. So, Heron in its current implementation running its GUI and 3 Nodes takes 22% of CPU load. This is still not ideal but is a consequence of the overhead of running multiple processes vs multiple threads. We believe that, given Heron’s latest optimisation, offering more control of system management to the user, the benefits of multi process applications outweigh this hit in system resources.

We have also increased the scope of the third example we provide in the paper and there we describe in detail how a full-scale experiment with 15 Nodes (which is the upper limit of number of Nodes usually required in most experiments) impacts CPU load.

Finally, we have added on Heron’s roadmap projects extra tasks focusing only on optimisation (profiling and using Numba for the time critical parts of the Heron code).

(3) I was also surprised to see that, despite being meant specifically to run on and connect diverse types of computer operating systems and being written purely in Python, the Heron Editor and GUI must be run on Windows. This seems like an unfortunate and unnecessary restriction, and it would be great to see the codebase adjusted to make it fully crossplatform-compatible.

This point was also mentioned by reviewer 2. This was a mistake on our part and has now been corrected in the paper. Heron (GUI and underlying communication functionality) can run on any machine that the underlying python libraries run, which is Windows, Linux (both for x86 and Arm architectures) and MacOS. We have tested it on Windows (10 and 11, both x64), Linux PC (Ubuntu 20.04.6, x64) and Raspberry Pi 4 (Debian GNU/Linux 12 (bookworm), aarch64). The Windows and Linux versions of Heron have undergone extensive debugging and all of the available Nodes (that are not OS specific) run on those two systems. We are in the process of debugging the Nodes’ functionality for RasPi. The MacOS version, although functional requires further work to make sure all of the basic Nodes are functional (which is not the case at the moment). We have also updated our manuscript (Multiple machines, operating systems and environments) to include the above information.

(4) Lastly, when I was running test experiments, sometimes one of the nodes, or part of the Heron editor itself would throw an exception or otherwise crash. Sometimes this left the Heron editor in a zombie state where some aspects of the GUI were responsive and others were not. It would be good to see a more graceful full shutdown of the program when part of it crashes or throws an exception, especially as this is likely to be common as people learn to use it. More problematically, in some of these cases, after closing or force quitting Heron, the TCP ports were not properly relinquished, and thus restarting Heron would run into an "address in use" error. Finding and killing the processes that were still using the ports is not something that is obvious, especially to a beginner, and it would be great to see Heron deal with this better. Ideally, code would be introduced to carefully avoid leaving ports occupied during a hard shutdown, and furthermore, when the address in use error comes up, it would be great to give the user some idea of what to do about it.

A lot of effort has been put into Heron to achieve graceful shut down of processes, especially when these run on different machines that do not know when the GUI process has closed. The code that is being suggested to avoid leaving ports open has been implemented and this works properly when processes do not crash (Heron is terminated by the user) and almost always when there is a bug in a process that forces it to crash. In the version of Heron available during the reviewing process there were bugs that caused the above behaviour (Node code hanging and leaving zombie processes) on MacOS systems. These have now been fixed. There are very seldom instances though, especially during Node development, that crashing processes will hang and need to be terminated manually. We have taken on board the reviewer’s comments that users should be made more aware of these issues and have also described this situation in the Debugging part of Heron’s documentation. There we explain the logging and other tools Heron provides to help users debug their own Nodes and how to deal with hanging processes.

Heron is still in alpha (usable but with bugs) and the best way to debug it and iron out all the bugs in all use cases is through usage from multiple users and error reporting (we would be grateful if the errors the reviewer mentions could be reported in Heron’s github Issues page). We are always addressing and closing any reported errors, since this is the only way for Heron to transition from alpha to beta and eventually to production code quality.

Overall I think that, with these improvements, this could be the beginning of a powerful and versatile new system that would enable flexible experiment design with a relatively low technical barrier to entry. I could see this system being useful to many different labs and fields.

We thank the reviewer for positive and supportive words and for the constructive feedbacks. We believe we have now addressed all the raised concerns.

**Reviewer #2 (Public Review):**
Summary:The authors provide an open-source graphic user interface (GUI) called Heron, implemented in Python, that is designed to help experimentalists to(1) design experimental pipelines and implement them in a way that is closely aligned with their mental schemata of the experiments,(2) execute and control the experimental pipelines with numerous interconnected hardware and software on a network.The former is achieved by representing an experimental pipeline using a Knowledge Graph and visually representing this graph in the GUI. The latter is accomplished by using an actor model to govern the interaction among interconnected nodes through messaging, implemented using ZeroMQ. The nodes themselves execute user-supplied code in, but not limited to, Python.Using three showcases of behavioral experiments on rats, the authors highlighted three benefits of their software design:(1) the knowledge graph serves as a self-documentation of the logic of the experiment, enhancing the readability and reproducibility of the experiment,(2) the experiment can be executed in a distributed fashion across multiple machines that each has a different operating system or computing environment, such that the experiment can take advantage of hardware that sometimes can only work on a specific computer/OS, a commonly seen issue nowadays,(3) he users supply their own Python code for node execution that is supposed to be more friendly to those who do not have a strong programming background.Strengths:(1) The software is light-weight and open-source, provides a clean and easy-to-use GUI,(2) The software answers the need of experimentalists, particularly in the field of behavioral science, to deal with the diversity of hardware that becomes restricted to run on dedicated systems.(3) The software has a solid design that seems to be functionally reliable and useful under many conditions, demonstrated by a number of sophisticated experimental setups.(4) The software is well documented. The authors pay special attention to documenting the usage of the software and setting up experiments using this software.Weaknesses:(1) While the software implementation is solid and has proven effective in designing the experiment showcased in the paper, the novelty of the design is not made clear in the manuscript. Conceptually, both the use of graphs and visual experimental flow design have been key features in many widely used softwares as suggested in the background section of the manuscript. In particular, contrary to the authors’ claim that only pre-defined elements can be used in Simulink or LabView, Simulink introduced MATLAB Function Block back in 2011, and Python code can be used in LabView since 2018. Such customization of nodes is akin to what the authors presented.

In the Heron manuscript we have provided an extensive literature review of existing systems from which Heron has borrowed ideas. We never wished to say that graphs and visual code is what sets Heron apart since these are technologies predating Heron by many years and implemented by a large number of software. We do not believe also that we have mentioned that LabView or Simulink can utilise only predefined nodes. What we have said is that in such systems (like LabView, Simulink and Bonsai) the focus of the architecture is on prespecified low level elements while the ability for users to author their own is there but only as an afterthought. The difference with Heron is that in the latter the focus is on the users developing their own elements. One could think of LabView style software as node-based languages (with low level visual elements like loops and variables) that also allow extra scripting while Heron is a graphical wrapper around python where nodes are graphical representations of whole processes. To our knowledge there is no other software that allows the very fast generation of graphical elements representing whole processes whose communication can also be defined graphically. Apart from this distinction, Heron also allows a graphical approach to writing code for processes that span different machines which again to our knowledge is a novelty of our approach and one of its strongest points towards ease of experimental pipeline creation (without sacrificing expressivity).

(2) The authors claim that the knowledge graph can be considered as a self-documentation of an experiment. I found it to be true to some extent. Conceptually it’s a welcoming feature and the fact that the same visualization of the knowledge graph can be used to run and control experiments is highly desirable (but see point 1 about novelty). However, I found it largely inadequate for a person to understand an experiment from the knowledge graph as visualized in the GUI alone. While the information flow is clear, and it seems easier to navigate a codebase for an experiment using this method, the design of the GUI does not make it a one-stop place to understand the experiment. Take the Knowledge Graph in Supplementary Figure 2B as an example, it is associated with the first showcase in the result section highlighting this self-documentation capability. I can see what the basic flow is through the disjoint graph where (1) one needs to press a key to start a trial, and (2) camera frames are saved into an avi file presumably using FFMPEG. Unfortunately, it is not clear what the parameters are and what each block is trying to accomplish without the explanation from the authors in the main text. Neither is it clear about what the experiment protocol is without the help of Supplementary Figure 2A.In my opinion, text/figures are still key to documenting an experiment, including its goals and protocols, but the authors could take advantage of the fact that they are designing a GUI where this information, with properly designed API, could be easily displayed, perhaps through user interaction. For example, in Local Network -> Edit IPs/ports in the GUI configuration, there is a good tooltip displaying additional information for the "password" entry. The GUI for the knowledge graph nodes can very well utilize these tooltips to show additional information about the meaning of the parameters, what a node does, etc, if the API also enforces users to provide this information in the form of, e.g., Python docstrings in their node template. Similarly, this can be applied to edges to make it clear what messages/data are communicated between the nodes. This could greatly enhance the representation of the experiment from the Knowledge graph.

In the first showcase example in the paper “Probabilistic reversal learning.

Implementation as self-documentation” we go through the steps that one would follow in order to understand the functionality of an experiment through Heron’s Knowledge Graph. The Graph is not just the visual representation of the Nodes in the GUI but also their corresponding code bases. We mention that the way Heron’s API limits the way a Node’s code is constructed (through an Actor based paradigm) allows for experimenters to easily go to the code base of a specific Node and understand its 2 functions (initialisation and worker) without getting bogged down in the code base of the whole Graph (since these two functions never call code from any other Nodes). Newer versions of Heron facilitate this easy access to the appropriate code by also allowing users to attach to Heron their favourite IDE and open in it any Node’s two scripts (worker and com) when they double click on the Node in Heron’s GUI. On top of this, Heron now (in the versions developed as answers to the reviewers’ comments) allows Node creators to add extensive comments on a Node but also separate comments on the Node’s parameters and input and output ports. Those can be seen as tooltips when one hovers over the Node (a feature that can be turned off or on by the Info button on every Node).

As Heron stands at the moment we have not made the claim that the Heron GUI is the full picture in the self-documentation of a Graph. We take note though the reviewer’s desire to have the GUI be the only tool a user would need to use to understand an experimental implementation. The solution to this is the same as the one described by the reviewer of using the GUI to show the user the parts of the code relevant to a specific Node without the user having to go to a separate IDE or code editor. The reason this has not been implemented yet is the lack of a text editor widget in the underlying gui library (DearPyGUI). This is in their roadmap for their next large release and when this exists we will use it to implement exactly the idea the reviewer is suggesting, but also with the capability to not only read comments and code but also directly edit a Node’s code (see Heron’s roadmap). Heron’s API at the moment is ideal for providing such a text editor straight from the GUI.

(3) The design of Heron was primarily with behavioral experiments in mind, in which highly accurate timing is not a strong requirement. Experiments in some other areas that this software is also hoping to expand to, for example, electrophysiology, may need very strong synchronization between apparatus, for example, the record timing and stimulus delivery should be synced. The communication mechanism implemented in Heron is asynchronous, as I understand it, and the code for each node is executed once upon receiving an event at one or more of its inputs. The paper, however, does not include a discussion, or example, about how Heron could be used to address issues that could arise in this type of communication. There is also a lack of information about, for example, how nodes handle inputs when their ability to execute their work function cannot keep up with the frequency of input events. Does the publication/subscription handle the queue intrinsically? Will it create problems in real-time experiments that make multiple nodes run out of sync? The reader could benefit from a discussion about this if they already exist, and if not, the software could benefit from implementing additional mechanisms such that it can meet the requirements from more types of experiments.

In order to address the above lack of explanation (that also the first reviewer pointed out) we expanded the third experimental example in the paper with three more sections. One focuses solely on explaining how in this example (which acquires and saves large amounts of data from separate Nodes running on different machines) one would be able to time align the different data packets generated in different Nodes to each other. The techniques described there are directly implementable on experiments where the requirements of synching are more stringent than the behavioural experiment we showcase (like in ephys experiments).

Regarding what happens to packages when the worker function of a Node is too slow to handle its traffic, this is mentioned in the paper (Code architecture paragraph): “Heron is designed to have no message buffering, thus automatically dropping any messages that come into a Node’s inputs while the Node’s worker function is still running.” This is also explained in more detail in Heron’s documentation. The reasoning for a no buffer system (as described in the documentation) is that for the use cases Heron is designed to handle we believe there is no situation where a Node would receive large amounts of data in bursts while very little data during the rest of the time (in which case a buffer would make sense). Nodes in most experiments will either be data intensive but with a constant or near constant data receiving speed (e.g. input from a camera or ephys system) or will have variable data load reception but always with small data loads (e.g. buttons). The second case is not an issue and the first case cannot be dealt with a buffer but with the appropriate code design, since buffering data coming in a Node too slow for its input will just postpone the inevitable crash. Heron’s architecture principle in this case is to allow these ‘mistakes’ (i.e. package dropping) to happen so that the pipeline continues to run and transfer the responsibility of making Nodes fast enough to the author of each Node. At the same time Heron provides tools (see the Debugging section of the documentation and the time alignment paragraph of the “Rats playing computer games” example in the manuscript) that make it easy to detect package drops and either correct them or allow them but also allow time alignment between incoming and outgoing packets. In the very rare case where a buffer is required Heron’s do-it-yourself logic makes it easy for a Node developer to implement their own Node specific buffer.

(4) The authors mentioned in "Heron GUI’s multiple uses" that the GUI can be used as an experimental control panel where the user can update the parameters of the different Nodes on the fly. This is a very useful feature, but it was not demonstrated in the three showcases. A demonstration could greatly help to support this claim.

As the reviewer mentions, we have found Heron’s GUI double role also as an experimental on-line controller a very useful capability during our experiments. We have expanded the last experimental example to also showcase this by showing how on the “Rats playing computer games” experiment we used the parameters of two Nodes to change the arena’s behaviour while the experiment was running, depending on how the subject was behaving at the time (thus exploring a much larger set of parameter combinations, faster during exploratory periods of our shaping protocols construction).

(5) The API for node scripts can benefit from having a better structure as well as having additional utilities to help users navigate the requirements, and provide more guidance to users in creating new nodes. A more standard practice in the field is to create three abstract Python classes, Source, Sink, and Transform that dictate the requirements for initialisation, work_function, and on_end_of_life, and provide additional utility methods to help users connect between their code and the communication mechanism. They can be properly docstringed, along with templates. In this way, the com and worker scripts can be merged into a single unified API. A simple example that can cause confusion in the worker script is the "worker_object", which is passed into the initialise function. It is unclear what this object this variable should be, and what attributes are available without looking into the source code. As the software is also targeting those who are less experienced in programming, setting up more guidance in the API can be really helpful. In addition, the self-documentation aspect of the GUI can also benefit from a better structured API as discussed in point 2 above.

The reviewer is right that using abstract classes to expose to users the required API would be a more standard practice. The reason we did not choose to do this was to keep Heron easily accessible to entry level Python programmers who do not have familiarity yet with object oriented programming ideas. So instead of providing abstract classes we expose only the implementation of three functions which are part of the worker classes but the classes themselves are not seen by the users of the API. The point about the users’ accessibility to more information regarding a few objects used in the API (the worker object for example) has been taken on board and we have now addressed this by type hinting all these objects both in the templates and more importantly in the automatically generated code that Heron now creates when a user chooses to create a Node graphically (a feature of Heron not present in the version available in the initial submission of this manuscript).

(6) The authors should provide more pre-defined elements. Even though the ability for users to run arbitrary code is the main feature, the initial adoption of a codebase by a community, in which many members are not so experienced with programming, is the ability for them to use off-the-shelf components as much as possible. I believe the software could benefit from a suite of commonly used Nodes.

There are currently 12 Node repositories in the Heron-repositories project on Github with more than 30 Nodes, 20 of which are general use (not implementing a specific experiment’ logic). This list will continue to grow but we fully appreciate the truth of the reviewer’s comment that adoption will depend on the existence of a large number of commonly used Nodes (for example Numpy, and OpenCV Nodes) and are working towards this goal.

(7) It is not clear to me if there is any capability or utilities for testing individual nodes without invoking a full system execution. This would be critical when designing new experiments and testing out each component.

There is no capability to run the code of an individual Node outside Heron’s GUI. A user could potentially design and test parts of the Node before they get added into a Node but we have found this to be a highly inefficient way of developing new Nodes. In our hands the best approach for Node development was to quickly generate test inputs and/or outputs using the “User Defined Function 1I 1O” Node where one can quickly write a function and make it accessible from a Node. Those test outputs can then be pushed in the Node under development or its outputs can be pushed in the test function, to allow for incremental development without having to connect it to the Nodes it would be connected in an actual pipeline. For example, one can easily create a small function that if a user presses a key will generate the same output (if run from a “User Defined Function 1I 1O” Node) as an Arduino Node reading some buttons. This output can then be passed into an experiment logic Node under development that needs to do something with this input. In this way during a Node development Heron allows the generation of simulated hardware inputs and outputs without actually running the actual hardware. We have added this way of developing Nodes also in our manuscript (Creating a new Node).

**Reviewer #3 (Public Review):**
Summary:The authors present a Python tool, Heron, that provides a framework for defining and running experiments in a lab setting (e.g. in behavioural neuroscience). It consists of a graphical editor for defining the pipeline (interconnected nodes with parameters that can pass data between them), an API for defining the nodes of these pipelines, and a framework based on ZeroMQ, responsible for the overall control and data exchange between nodes. Since nodes run independently and only communicate via network messages, an experiment can make use of nodes running on several machines and in separate environments, including on different operating systems.Strengths:As the authors correctly identify, lab experiments often require a hodgepodge of separate hardware and software tools working together. A single, unified interface for defining these connections and running/supervising the experiment, together with flexibility in defining the individual subtasks (nodes) is therefore a very welcome approach. The GUI editor seems fairly intuitive, and Python as an accessible programming environment is a very sensible choice. By basing the communication on the widely used ZeroMQ framework, they have a solid base for the required non-trivial coordination and communication. Potential users reading the paper will have a good idea of how to use the software and whether it would be helpful for their own work. The presented experiments convincingly demonstrate the usefulness of the tool for realistic scientific applications.Weaknesses:(1) In my opinion, the authors somewhat oversell the reproducibility and "selfdocumentation" aspect of their solution. While it is certainly true that the graph representation gives a useful high-level overview of an experiment, it can also suffer from the same shortcomings as a "pure code" description of a model - if a user gives their nodes and parameters generic/unhelpful names, reading the graph will not help much.

This is a problem that to our understanding no software solution can possibly address. Yet having a visual representation of how different inputs and outputs connect to each other we argue would be a substantial benefit in contrast to the case of “pure code” especially when the developer of the experiment has used badly formatted variable names.

(2) Making the link between the nodes and the actual code is also not straightforward, since the code for the nodes is spread out over several directories (or potentially even machines), and not directly accessible from within the GUI.

This is not accurate. The obligatory code of a Node always exists within a single folder and Heron’s API makes it rather cumbersome to spread scripts relating to a Node across separate folders. The Node folder structure can potentially be copied over different machines but this is why Heron is tightly integrated with git practices (and even politely asks the user with popup windows to create git repositories of any Nodes they create whilst using Heron’s automatic Node generator system). Heron’s documentation is also very clear on the folder structure of a Node which keeps the required code always in the same place across machines and more importantly across experiments and labs. Regarding the direct accessibility of the code from the GUI, we took on board the reviewers’ comments and have taken the first step towards correcting this. Now one can attach to Heron their favourite IDE and then they can double click on any Node to open its two main scripts (com and worker) in that IDE embedded in whatever code project they choose (also set in Heron’s settings windows). On top of this, Heron now allows the addition of notes both for a Node and for all its parameters, inputs and outputs which can be viewed by hovering the mouse over them on the Nodes’ GUIs. The final step towards GUI-code integration will be to have a Heron GUI code editor but this is something that has to wait for further development from Heron’s underlying GUI library DearPyGUI.

(3) The authors state that "[Heron’s approach] confers obvious benefits to the exchange and reproducibility of experiments", but the paper does not discuss how one would actually exchange an experiment and its parameters, given that the graph (and its json representation) contains user-specific absolute filenames, machine IP addresses, etc, and the parameter values that were used are stored in general data frames, potentially separate from the results. Neither does it address how a user could keep track of which versions of files were used (including Heron itself).

Heron’s Graphs, like any experimental implementation, must contain machine specific strings. These are accessible either from Heron’s GUI when a Graph json file is opened or from the json file itself. Heron in this regard does not do anything different to any other software, other than saving the graphs into human readable json files that users can easily manipulate directly.

Heron provides a method for users to save every change of the Node parameters that might happen during an experiment so that it can be fully reproduced. The dataframes generated are done so in the folders specified by the user in each of the Nodes (and all those paths are saved in the json file of the Graph). We understand that Heron offers a certain degree of freedom to the user (Heron’s main reason to exist is exactly this versatility) to generate data files wherever they want but makes sure every file path gets recorded for subsequent reproduction. So, Heron behaves pretty much exactly like any other open source software. What we wanted to focus on as the benefits of Heron on exchange and reproducibility was the ability of experimenters to take a Graph from another lab (with its machine specific file paths and IP addresses) and by examining the graphical interface of it to be able to quickly tweak it to make it run on their own systems. That is achievable through the fact that a Heron experiment will be constructed by a small amount of Nodes (5 to 15 usually) whose file paths can be trivially changed in the GUI or directly in the json file while the LAN setup of the machines used can be easily reconstructed from the information saved in the secondary GUIs.

Where Heron needs to improve (and this is a major point in Heron’s roadmap) is the need to better integrate the different saved experiments with the git versions of Heron and the Nodes that were used for that specific save. This, we appreciate is very important for full reproducibility of the experiment and it is a feature we will soon implement. More specifically users will save together with a graph the versions of all the used repositories and during load the code base utilised will come from the recorded versions and not from the current head of the different repositories. This is a feature that we are currently working on now and as our roadmap suggests will be implemented by the release of Heron 1.0.

(4) Another limitation that in my opinion is not sufficiently addressed is the communication between the nodes, and the effect of passing all communications via the host machine and SSH. What does this mean for the resulting throughput and latency - in particular in comparison to software such as Bonsai or Autopilot? The paper also states that "Heron is designed to have no message buffering, thus automatically dropping any messages that come into a Node’s inputs while the Node’s worker function is still running."- it seems to be up to the user to debug and handle this manually?

There are a few points raised here that require addressing. The first is Heron’s requirement to pass all communication through the main (GUI) machine. We understand (and also state in the manuscript) that this is a limitation that needs to be addressed. We plan to do this is by adding to Heron the feature of running headless (see our roadmap). This will allow us to run whole Heron pipelines in a second machine which will communicate with the main pipeline (run on the GUI machine) with special Nodes. That will allow experimenters to define whole pipelines on secondary machines where the data between their Nodes stay on the machine running the pipeline. This is an important feature for Heron and it will be one of the first features to be implemented next (after the integration of the saving system with git).

The second point is regarding Heron’s throughput latency. In our original manuscript we did not have any description of Heron’s capabilities in this respect and both other reviewers mentioned this as a limitation. As mentioned above, we have now addressed this by adding a section to our third experimental example that fully describes how much CPU is required to run a full experimental pipeline running on two machines and utilising also non python code executables (a Unity game). This gives an overview of how heavy pipelines can run on normal computers given adequate optimisation and utilising Heron’s feature of forcing some Nodes to run their Worker processes on a specific core. At the same time, Heron’s use of 0MQ protocol makes sure there are no other delays or speed limitations to message passing. So, message passing within the same machine is just an exchange of memory pointers while messages passing between different machines face the standard speed limitations of the Local Access Network’s ethernet card speeds.

Finally, regarding the message dropping feature of Heron, as mentioned above this is an architectural decision given the use cases of message passing we expect Heron to come in contact with. For a full explanation of the logic here please see our answer to the 3rd comment by Reviewer 2.

(5) As a final comment, I have to admit that I was a bit confused by the use of the term "Knowledge Graph" in the title and elsewhere. In my opinion, the Heron software describes "pipelines" or "data workflows", not knowledge graphs - I’d understand a knowledge graph to be about entities and their relationships. As the authors state, it is usually meant to make it possible to "test propositions against the knowledge and also create novel propositions" - how would this apply here?

We have described Heron as a Knowledge Graph instead of a pipeline, data workflow or computation graph in order to emphasise Heron’s distinct operation in contrast to what one would consider a standard pipeline and data workflow generated by other visual based software (like LabView and Bonsai). This difference exists on what a user should think of as the base element of a graph, i.e. the Node. In all other visual programming paradigms, the Node is defined as a low-level computation, usually a language keyword, language flow control or some simple function. The logic in this case is generated by composing together the visual elements (Nodes). In Heron the Node is to be thought of as a process which can be of arbitrary complexity and the logic of the graph is composed by the user both within each Node and by the way the Nodes are combined together. This is an important distinction in Heron’s basic operation logic and it is we argue the main way Heron allows flexibility in what can be achieved while retaining ease of graph composition (by users defining their own level of complexity and functionality encompassed within each Node). We have found that calling this approach a computation graph (which it is) or a pipeline or data workflow would not accentuate this difference. The term Knowledge Graph was the most appropriate as it captures the essence of variable information complexity (even in terms of length of shortest string required) defined by a Node.

**Recommendations for the authors:**

**Reviewer #1 (Recommendations For The Authors):**
- No buffering implies dropped messages when a node is busy. It seems like this could be very problematic for some use cases...

This is a design principle of Heron. We have now provided a detailed explanation of the reasoning behind it in our answer to Reviewer 2 (Paragraph 3) as well as in the manuscript.

- How are ssh passwords stored, and is it secure in some way or just in plain text?

For now they are plain text in an unencrypted file that is not part of the repo (if one gets Heron from the repo). Eventually, we would like to go to private/public key pairs but this is not a priority due to the local nature of Heron’s use cases (all machines in an experiment are expected to connect in a LAN).

Minor notes / copyedits:- Figure 2A: right and left seem to be reversed in the caption.

They were. This is now fixed.

- Figure 2B: the text says that proof of life messages are sent to each worker process but in the figure, it looks like they are published by the workers? Also true in the online documentation.

The Figure caption was wrong. This is now fixed.

- psutil package is not included in the requirements for GitHub

We have now included psutil in the requirements.

- GitHub readme says Python >= 3.7 but Heron will not run as written without python >= 3.9 (which is alluded to in the paper)

The new Heron updates require Python 3.11. We have now updated GitHub and the documentation to reflect this.

- The paper mentions that the Heron editor must be run on Windows, but this is not mentioned in the Github readme.

This was an error in the manuscript that we have now corrected.

- It’s unclear from the readme/manual how to remove a node from the editor once it’s been added.

We have now added an X button on each Node to complement the Del button on the keyboard (for MacOS users that do not have this button most of the times).

- The first example experiment is called the Probabilistic Reversal Learning experiment in text, but the uncertainty experiment in the supplemental and on GitHub.

We have now used the correct name (Probabilistic Reversal Learning) in both the supplemental material and on GitHub

- Since Python >= 3.9 is required, consider using fstrings instead of str.format for clarity in the codebase

Thank you for the suggestion. Latest Heron development has been using f strings and we will do a refactoring in the near future.

- Grasshopper cameras can run on linux as well through the spinnaker SDK, not just Windows.

Fixed in the manuscript.

- Figure 4: Square and star indicators are unclear.

Increased the size of the indicators to make them clear.

- End of page 9: "an of the self" presumably a typo for "off the shelf"?

Corrected.

- Page 10 first paragraph. "second root" should be "second route"

Corrected.

- When running Heron, the terminal constantly spams Blowfish encryption deprecation warnings, making it difficult to see the useful messages.

The solution to this problem is to either update paramiko or install Heron through pip. This possible issue is mentioned in the documentation.

- Node input /output hitboxes in the GUI are pretty small. If they could be bigger it would make it easier to connect nodes reliably without mis-clicks.

We have redone the Node GUI, also increasing the size of the In/Out points.

**Reviewer #2 (Recommendations For The Authors):**
(1) There are quite a few typos in the manuscript, for example: "one can accessess the code", "an of the self", etc.

Thanks for the comment. We have now screened the manuscript for possible typos.

(2) Heron’s GUI can only run on Windows! This seems to be the opposite of the key argument about the portability of the experimental setup.

As explained in the answers to Reviewer 1, Heron can run on most machines that the underlying python libraries run, i.e. Windows and Linux (both for x86 and Arm architectures). We have tested it on Windows (10 and 11, both x64), Linux PC (Ubuntu 20.04.6, x64) and Raspberry Pi 4 (Debian GNU/Linux 12 (bookworm), aarch64). We have now revised the manuscript and the GitHub repo to reflect this.

(3) Currently, the output is displayed along the left edge of the node, but the yellow dot connector is on the right. It would make more sense to have the text displayed next to the connectors.

We have redesigned the Node GUI and have now placed the Out connectors on the right side of the Node.

(4) The edges are often occluded by the nodes in the GUI. Sometimes it leads to some confusion, particularly when the number of nodes is large, e.g., Fig 4.

This is something that is dependent on the capabilities of the DearPyGUI module. At the moment there is no way to control the way the edges are drawn.

**Reviewer #3 (Recommendations For The Authors):**
A few comments on the software and the documentation itself:- From a software engineering point of view, the implementation seems to be rather immature. While I get the general appeal of "no installation necessary", I do not think that installing dependencies by hand and cloning a GitHub repository is easier than installing a standard package.

We have now added a pip install capability which also creates a Heron command line command to start Heron with.

-The generous use of global variables to store state (minor point, given that all nodes run in different processes), boilerplate code that each node needs to repeat, and the absence of any kind of automatic testing do not give the impression of a very mature software (case in point: I had to delete a line from editor.py to be able to start it on a non-Windows system).

As mentioned, the use of global variables in the worker scripts is fine partly due to the multi process nature of the development and we have found it is a friendly approach to Matlab users who are just starting with Python (a serious consideration for Heron). Also, the parts of the code that would require a singleton (the Editor for example) are treated as scripts with global variables while the parts that require the construction of objects are fully embedded in classes (the Node for example). A future refactoring might make also all the parts of the code not seen by the user fully object oriented but this is a decision with pros and cons needing to be weighted first.

Absence of testing is an important issue we recognise but Heron is a GUI app and nontrivial unit tests would require some keystroke/mouse movement emulator (like QTest of pytest-qt for QT based GUIs). This will be dealt with in the near future (using more general solutions like PyAutoGUI) but it is something that needs a serious amount of effort (quite a bit more that writing unit tests for non GUI based software) and more importantly it is nowhere as robust as standard unit tests (due to the variable nature of the GUI through development) making automatic test authoring an almost as laborious a process as the one it is supposed to automate.

- From looking at the examples, I did not quite see why it is necessary to write the ..._com.py scripts as Python files, since they only seem to consist of boilerplate code and variable definitions. Wouldn’t it be more convenient to represent this information in configuration files (e.g. yaml or toml)?

The com is not a configuration file, it is a script that launches the communication process of the Node. We could remove the variable definitions to a separate toml file (which then the com script would have to read). The pros and cons of such a set up should be considered in a future refactoring.

Minor comments for the paper:- p.7 (top left): "through its return statement" - the worker loop is an infinite loop that forwards data with a return statement?

This is now corrected. The worker loop is an infinite loop and does not return anything but at each iteration pushes data to the Nodes output.

- p.9 (bottom right): "of the self" → "off-the-shelf"

Corrected.

- p.10 (bottom left): "second root" → "second route"

Corrected.

- Supplementary Figure 3: Green start and square seem to be swapped (the green star on top is a camera image and the green star on the bottom is value visualization - inversely for the green square).

The star and square have been swapped around.

- Caption Supplementary Figure 4 (end): "rashes to receive" → "rushes to receive"

Corrected.